# Proteostasis by STUB1/HSP70 complex controls sensitivity to androgen receptor targeted therapy in advanced prostate cancer

Chengfei Liu[1], Wei Lou[1], Joy C. Yang[1], Liangren Liu[2], Cameron M. Armstrong[1], Alan P. Lombard[1], Ruining Zhao[3], Onika D.V. Noel[1], Clifford G. Tepper [4,5], Hong-Wu Chen[4,5,6], Marc Dall'Era[1,5], Christopher P. Evans[1,5] & Allen C. Gao[1,5,6]

Protein homeostasis (proteostasis) is a potential mechanism that contributes to cancer cell survival and drug resistance. Constitutively active androgen receptor (AR) variants confer anti-androgen resistance in advanced prostate cancer. However, the role of proteostasis involved in next generation anti-androgen resistance and the mechanisms of AR variant regulation are poorly defined. Here we show that the ubiquitin-proteasome-system (UPS) is suppressed in enzalutamide/abiraterone resistant prostate cancer. AR/AR-V7 proteostasis requires the interaction of E3 ubiquitin ligase STUB1 and HSP70 complex. STUB1 disassociates AR/AR-V7 from HSP70, leading to AR/AR-V7 ubiquitination and degradation. Inhibition of HSP70 significantly inhibits prostate tumor growth and improves enzalutamide/abiraterone treatments through AR/AR-V7 suppression. Clinically, HSP70 expression is upregulated and correlated with AR/AR-V7 levels in high Gleason score prostate tumors. Our results reveal a novel mechanism of anti-androgen resistance via UPS alteration which could be targeted through inhibition of HSP70 to reduce AR-V7 expression and overcome resistance to AR-targeted therapies.

[1] Department of Urology, University of California Davis, Davis 95817 CA, USA. [2] Department of Urology, West China Hospital, Sichuan University, Chengdu 610041, China. [3] Department of Urology, General Hospital of Ningxia Medical University, Yinchuan 750100, China. [4] Department of Biochemistry and Molecular Medicine, University of California Davis, Davis 95817 CA, USA. [5] UC Davis Comprehensive Cancer Center, University of California Davis, Davis 95817 CA, USA. [6] VA Northern California Health Care System, Sacramento 95655 CA, USA. Correspondence and requests for materials should be addressed to A.C.G. (email: acgao@ucdavis.edu)

Proteomic equilibrium including protein folding, trafficking, maturation, and degradation controls mammalian cell biological function and maintains physiological environment stabilization. Protein homeostasis (proteostasis) is regulated through a comprehensive network, including molecular chaperone proteins, the ubiquitin–proteasome system, and the autophagy system[1–5]. Imbalanced proteostasis disrupts protein clearance and increases abnormal deposition of protein aggregates which facilitates cancer cell survival and progression. Thus, overexpression of oncogenic proteins mediated by proteostasis is a potential mechanism that contributes to drug resistance in cancer cells. Understanding the mechanisms of protein post-translational regulation in order to find strategies to correct proteostasis-imbalance in anti-androgen resistant prostate cancer is warranted.

Enzalutamide and abiraterone are the second-generation anti-androgen drugs approved for the treatment of castration-resistant prostate cancer (CRPC). Even though they are effective at first, resistance to both drugs occurs frequently. Considerable evidence from both clinical and experimental studies demonstrate that truncated androgen receptor (AR) variants, particularly AR-V7, plays vital roles in promoting CRPC progression during androgen deprivation therapy and in the induction of resistance to enzalutamide and abiraterone therapy[6–9]. Rearrangements that alter AR gene structure and splicing patterns have been described in prostate cancer cell lines, and xenografts which suggests the origin of AR-V7 might be derived from intragenic AR gene rearrangements or premature translation termination by aberrant mRNA splicing[10–12]. However, post-translational regulation of AR-V7 and the mechanisms of AR-V7 proteostasis have not been fully explored.

The chaperone protein family, including heat shock proteins (HSPs), regulates the activity and stability of many oncogenes that control cancer cell survival and progression[3,13–15]. The HSP70s family, including stress inducible member HSP70 (HSPA1A/HSPA1B) and constitutively expressed member HSC70 (HSPA8), plays important roles for protein maturation and correct folding in cancer cell signal transduction and regulation[16–18]. STUB1 is a co-chaperone protein and functional E3 ubiquitin ligase that links HSP70's polypeptide-binding activity to the ubiquitin proteasome system. HSP70 interacts with STUB1 and controls protein stabilization. Binding of STUB1 to HSP70 can halt the proper folding of HSP70 substrate proteins and concomitantly facilitate the U-box-dependent ubiquitination of HSP70-bound substrates[19–21]. As AR's co-chaperone protein, HSP70 assists the folding and maturation of AR protein[22–24]. However, understanding of the interaction among AR-V7, HSP70, and STUB1 in next generation anti-androgen resistance remains elusive.

In the present study, we discover that the ubiquitin-mediated proteolysis pathway and proteasome activity are suppressed in enzalutamide and abiraterone-resistant prostate cancer cells which stabilizes AR-V7 protein in these cells through ubiquitin–proteasome alteration. The STUB1/HSP70 complex regulates full length AR (AR-FL) and AR variant proteostasis which confers next generation anti-androgen resistance. HSP70 inhibition significantly disrupts AR and AR-V7 gene programs and re-sensitizes resistant cells to enzalutamide and abiraterone treatment both in vitro and in vivo. Notably, the levels of HSP70 are correlated with AR-V7 in tumors from patients with high Gleason scores. These results suggest that targeting the proteostasis pathway through inhibiting HSP70 might be a valuable strategy to overcome next generation anti-androgen resistance and improve drug therapy in CRPC patients.

## Results

### UPS suppressing confers AR-FL/AR-V7 protein stabilization.
Enzalutamide and abiraterone-resistant CWR22Rv1 and C4-2B

MDVR cells express both AR-FL and AR-V7 as demonstrated by RNA transcriptome sequencing. The AR mRNA splice junction was analyzed by Integrative Genomics Viewer (IGV) 2.4. C4-2B MDVR and CWR22Rv1 cells showed abundant splice junctions between AR exon3 and exon4 (Fig. 1a). Among the products derived from these splice junctions are AR-V1, AR-V3, AR-V7, and AR-V9, with AR-V7 being the most abundant AR variant in both C4-2B MDVR (depth 22 reads) and CWR22Rv1 cells (depth 111 reads). Both C4-2B MDVR and CWR22Rv1 cells express higher AR-FL and AR-V7 mRNA and protein levels compared to C4-2B cells as confirmed by real-time reverse transcription-PCR (qRT-PCR) and Western blot (Fig. 1b). Consistently, C4-2B MDVR and CWR22Rv1 xenograft tumors express significantly higher levels of AR-V7 protein compared to C4-2B parental xenograft tumors as measured by IHC (Fig. 1c). To understand the potential mechanisms that may be involved in overexpression of AR-V7 protein in enzalutamide-resistant cells, we determined whether AR-V7 protein stabilization was altered in C4-2B MDVR cells. C4-2B parental and C4-2B MDVR cells were treated with cycloheximide (CHX), AR-V7 and AR-FL protein levels were examined at different time points. Both AR-V7 and AR-FL proteins were more stable in C4-2B MDVR cells compared to the C4-2B parental cells (Fig. 1d). Half-life of AR-V7 protein in C4-2B cells was ~2 h, while AR-V7 half-life in C4-2B MDVR cells was significantly increased (>8 h). Half-life of AR-FL protein was also increased in C4-2B MDVR cells. We next analyzed global gene microarray data from C4-2B parental, C4-2B MDVR, and C4-2B AbiR cells by ingenuity pathway analysis (IPA) and GSEA. The protein ubiquitination pathway was the second highest altered pathway in enzalutamide-resistant cells ($p = 1.93 \times 10^{-7}$) (Fig. 1e). GSEA revealed that the ubiquitin-mediated proteolysis pathway was significantly suppressed in C4-2B MDVR and C4-2B AbiR cells ($p < 0.05$). The results were also confirmed in other independent enzalutamide-resistant cells (LNCaP EnzaR and CWR-R1 EnzaR)[25] and LuCaP castration-resistant (LuCaP CR)[26] patient-derived xenograft (PDX) models overexpressing AR-V7 (Fig. 1f and Supplementary Fig.1a). The ubiquitin-mediated proteolysis pathway was significantly suppressed in LNCaP EnzaR ($p < 0.001$), CWR-R1 EnzaR ($p < 0.001$) cells, and LuCaP CR PDX tumor models ($p < 0.001$). Further proteasome activity fluorometric assay revealed that resistant C4-2B AbiR, C4-2B MDVR, and CWR22Rv1 cells produced significantly lower proteasome activity compared to C4-2B cells (Fig. 1g). Moreover, AR and AR-V7 ubiquitination were significantly suppressed in all resistant cells (Fig. 1h). Taken together, the results suggest that AR-V7 is overexpressed in resistant prostate cancer cells via enhanced protein stability through ubiquitin proteasome system alteration.

### STUB1/HSP70 complex regulates AR variants protein expression.
Heatmap analysis of global gene microarray data from enzalutamide/abiraterone-resistant cells and LuCaP CR tumors showed that the core enrichment genes involved in protein ubiquitination were downregulated. One of the most significantly downregulated genes in the resistant cells and LuCaP CR tumors was E3 ubiquitin ligase STUB1 (Fig. 2a and Supplementary Fig. 1b). We found that STUB1 expression was significantly reduced in AR-V7 positive prostate cancer cells, such as C4-2B MDVR, C4-2B AbiR, CWR22Rv1, and VCaP cells compared to the AR-V7 negative LNCaP cells (Fig. 2b). Next, we determined whether STUB1 and HSP70 (a STUB1-binding protein) bind to AR-V7 or AR-FL by co-immunoprecipitation (Co-IP) assays in the 293 cell system. We first confirmed that STUB1 binds with HSP70 (Supplemental Fig. 1c top). Then we found that AR-V1, AR-V3, AR-V7, AR-V9, AR-V12 (AR567es), and AR-FL directly bind to

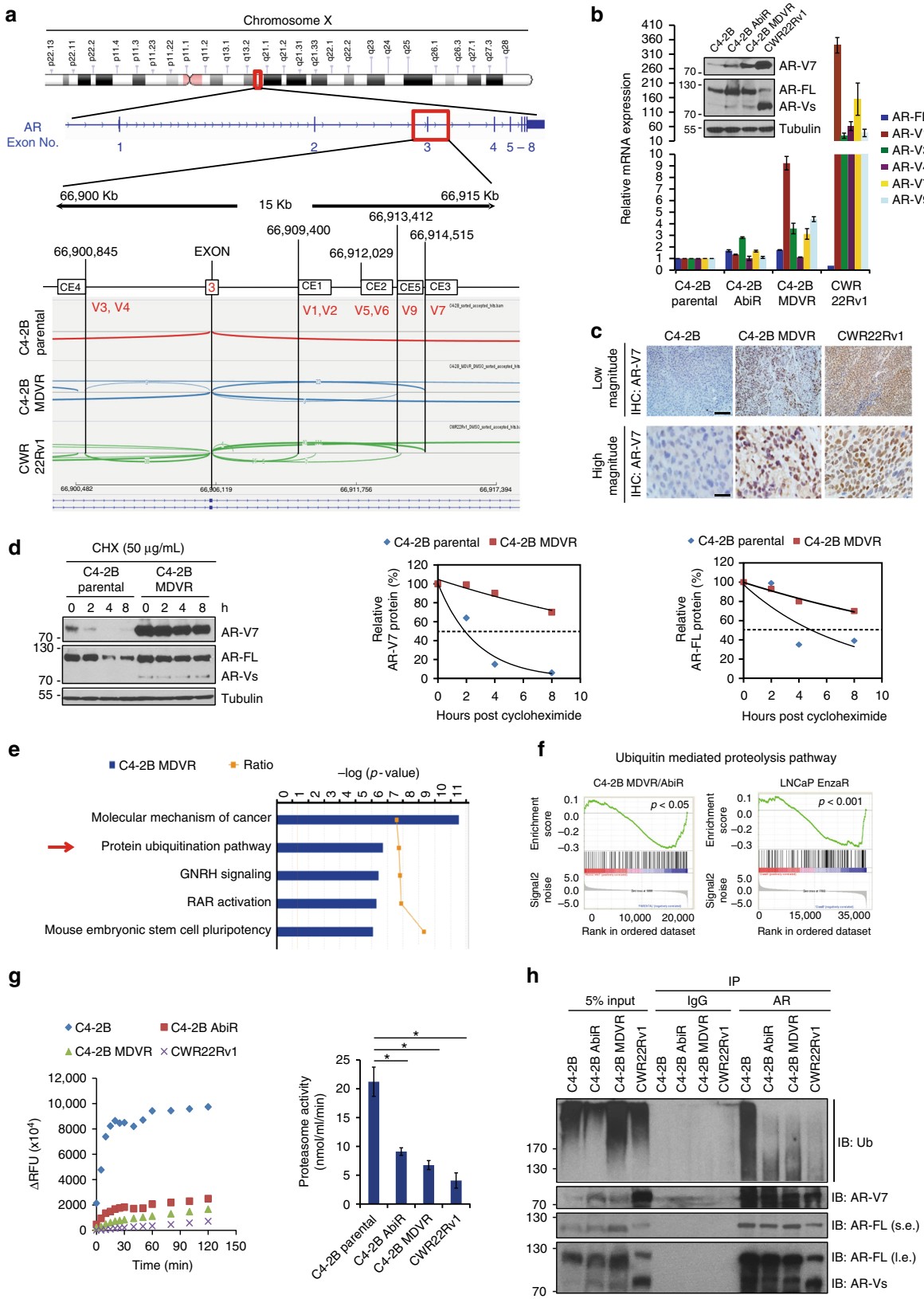

STUB1 and HSP70 (Fig. 2c, Supplementary Fig. 1c middle and bottom). Similar results showed that STUB1 and HSP70 bind to endogenous AR-FL and AR variants in both C4-2B MDVR and CWR22Rv1 cells (Supplementary Fig. 1d).

To further investigate the effect of STUB1 on regulation of AR-V7 expression, we transiently transfected STUB1 into C4-2B MDVR and CWR22Rv1 cells and found that overexpression of STUB1 significantly suppressed AR-V7 protein expression but

**Fig. 1** UPS suppressing confers AR-FL/AR-V7 protein stabilization. **a** RNA transcriptome sequence data from C4-2B parental, C4-2B MDVR, and CWR22Rv1 cells were viewed by IGV2.4 and AR splicing conjunction around exon3 was analyzed by sashimi plot. **b** Total RNA from C4-2B parental, C4-2B MDVR, and CWR22Rv1 cells was extracted and mRNA levels of AR-FL, AR-V1, AR-V3, AR-V7, AR-V9, and AR-V12 were examined by qRT-PCR. AR-V7 and AR-FL protein level in C4-2B parental, C4-2B AbiR, C4-2B MDVR, and CWR22Rv1 cells were examined by western blot. **c** AR-V7 immunohistochemistry staining of the tumor sections isolated from the C4-2B parental, C4-2B MDVR, and CWR22Rv1 xenograft tumors. Scale bar 100 μm (low) and 20 μm (high). **d** C4-2B parental and C4-2B MDVR cells were treated with 50 μg/mL cycloheximide, total cell lysates were collected at 0, 2, 4, and 8 h after treatment. AR-V7 and AR-FL was examined by western blot and the half-life of AR-V7 was calculated. **e** Global gene microarray data from C4-2B parental and C4-2B MDVR cells was analyzed by IPA, the top five canonical pathways were altered in C4-2B MDVR cells. **f** Global gene microarray data from C4-2B parental, C4-2B MDVR, and C4-2B AbiR cells, LNCaP and LNCaP EnzaR cells were analyzed by GSEA, ubiquitin- mediated proteolysis gene set was enriched in resistant cells. **g** C4-2B, C4-2B MDVR, C4-2B AbiR, and CWR22Rv1 cells were harvested and proteasome activity was determined by proteasome Activity Fluorometric assay kit. ΔRFU was monitored at different time points (left) and the proteasome activity was quantified (right). **h** Whole cell lysates from C4-2B, C4-2B MDVR, C4-2B AbiR, and CWR22Rv1 cells were harvested and immunoprecipitated with AR antibody and blotted with anti-Ubiquitin, AR-V7, and AR antibodies. *$p < 0.05$. Results are the mean of three independent experiments (±s.d.). Statistical analysis was performed using two-tailed Student's $t$-test. AR-FL full-length AR, AR-Vs AR-Variants, Ub Ubiquitin, s.e. short exposure, l.e. long exposure

did not affect mRNA expression (Fig. 2d and Supplementary Fig. 2a). STUB1 suppressed while HSP70 increased AR-V7 transcriptional activity in C4-2B cells (Supplementary Fig. 2b). We then investigated whether STUB1 affects AR-V7 and AR-FL protein stability. As shown in Fig. 2e and Supplementary Fig. 2c, STUB1 overexpression significantly shortened the half-life of AR-V7 protein (around 2 h) in CWR22Rv1 and C4-2B MDVR cells compared to the vector-transfected controls (around 8 h). The AR-FL protein half-life was also shortened by STUB1 over-expression in both CWR22Rv1 and C4-2B MDVR cells. To determine whether decreased AR-V7 protein expression by STUB1 expression is mediated through the ubiquitin–proteasome pathway, proteasome inhibitor MG132 was added into C4-2B or 293 cells transfected with AR-V7 and STUB1. STUB1 decreased AR-V7 expression, while addition of MG132 blunted the STUB1 effects (Fig. 2f and Supplementary Fig. 2d). Furthermore, STUB1 significantly induced AR-V7 ubiquitination (Fig. 2g), suggesting that STUB1-mediated AR-V7 protein degradation is through the induction of AR-V7 ubiquitination. In addition, HSP70 formed complexes with AR-V7/AR-FL, while STUB1 disassociated AR-V7/AR-FL from HSP70 binding (Fig. 2h and Supplementary Fig. 2e). These results indicate that the STUB1/HSP70 complex regulates AR-V7 and AR-FL protein expression through the ubiquitin–proteasome pathway. Since AR-V7 is associated with enzalutamide resistance, we tested if STUB1 affects cell sensitivity to enzalutamide. CWR22Rv1 and C4-2B MDVR cells expressing lower level of STUB1 were transiently transfected with STUB1 and subsequently treated with enzalutamide or abiraterone for 3 days. STUB1 overexpression re-sensitized drug- resistant cells to enzalutamide or abiraterone treatment (Fig. 2i). These results suggest that STUB1 modulates cells sensitivity to enzalutamide/abiraterone treatment possibly through downregulation of AR-V7 protein expression.

**STUB1/HSP70 complex regulates sensitivity to anti-androgens**. As a co-chaperone protein of STUB1, HSP70 assists oncogenic proteins folding and maturation. We found that HSP70 but not HSP90 (HSP90AA1) was overexpressed in enzalutamide-resistant prostate cancer cells and LuCaP CR tumors (Fig. 3a and Supplementary Fig. 2f). HSP70 overexpression blocked STUB1 and AR-V7 binding as demonstrated by Co-IP (Fig. 3b left) and decreased AR-V7 ubiquitination (Fig. 3b right). To elucidate whether HSP70 is also involved in resistance to enzalutamide and abiraterone, HSP70 was knocked down by two independent siRNA in CWR22Rv1 and C4-2B MDVR cells. Following siRNA transfection, the cells were treated with enzalutamide or abiraterone. Knockdown of HSP70 significantly re-sensitized both CWR22Rv1 and C4-2B MDVR cells to enzalutamide (Fig. 3c) and

abiraterone treatments (Supplementary Fig. 3a). Notably, knockdown of HSP70 significantly decreased AR-V7 as well as AR-FL expression in both CWR22Rv1 and C4-2B MDVR cells (Fig. 3d left) and consequently suppressed PSA luciferase activity (Fig. 3d right). To further demonstrate HSP70-conferred enzalutamide and abiraterone resistance is through the AR/AR-V7 regulation, human normal fibroblast cells IMR90 and immortalized prostate epithelial cells PZ-HPV7 were used. As shown in Supplementary Fig. 3b, IMR90 and PZ-HPV7 cells are AR and AR-V7 negative cells and they expressed significantly lower levels of HSP70 compared to C4-2B MDVR cells. Knockdown of HSP70 in IMR90 and PZ-HPV7 did not affect enzalutamide and abiraterone sensitivity (Supplementary Fig. 3c-d). These results suggest that HSP70 confers resistance to next generation anti-androgen treatments through AR-V7 regulation, indicating that HSP70 could serve as a therapeutic target.

Apoptozole (APO)[27] and Ver155008 (VER)[28] are HSP70 inhibitors that function through binding to the HSP70 active pocket and suppressing its ATPase activity (Supplementary Fig. 3e-f). We determined that APO and VER significantly suppressed the growth of the resistant prostate cancer cells in a dose-dependent manner but had moderate effects on PZ-HPV7 and no effects on IMR90 cells (Fig. 3e). APO and VER significantly reduced AR-V7 protein expression in both C4-2B MDVR and CWR22Rv1 cells (Fig. 3f). Both APO and VER suppressed the PSA luciferase activity in CWR22Rv1 and C4-2B MDVR cells (Supplementary Fig. 3g). Additionally, APO and VER suppressed DHT and AR-V7-induced PSA luciferase activity, especially the HSP70- induced AR-V7 transcriptional activity in C4-2B cells. However, enzalutamide only suppressed DHT-induced PSA luciferase activity but not HSP70-induced AR-V7 transcriptional activity in C4-2B cells (Supplementary Fig. 3h). Glucocorticoid receptor (GR) has been linked to enzalutamide resistance previously[29]. We found the STUB1/HSP70 complex also regulated GR activity. APO and VER treatment or STUB1 overexpression significantly suppressed GR activity (Supplementary Fig. 3i-j). We next determined if APO and VER affect mRNA expression of AR variants and their targets genes in C4-2B MDVR cells (Supplementary Fig. 4a-g). Intriguingly, APO slightly decreased the levels of AR-V7 mRNA, but significantly decreased the levels of AR-FL, AR-V1, AR-V3, and AR-V9 mRNA. VER significantly decreased the levels of AR-V3 and AR-V9 but not the levels of AR-FL and AR-V7 mRNA. These results suggested that APO and VER alter AR/AR variants expression at both mRNA and protein levels. However, addition of proteasome inhibitor MG132 largely rescued APO and VER effects on AR/AR variants suppression (Supplementary Fig. 4f), suggesting AR and its variants suppression by APO and VER is largely due to the protein degradation.

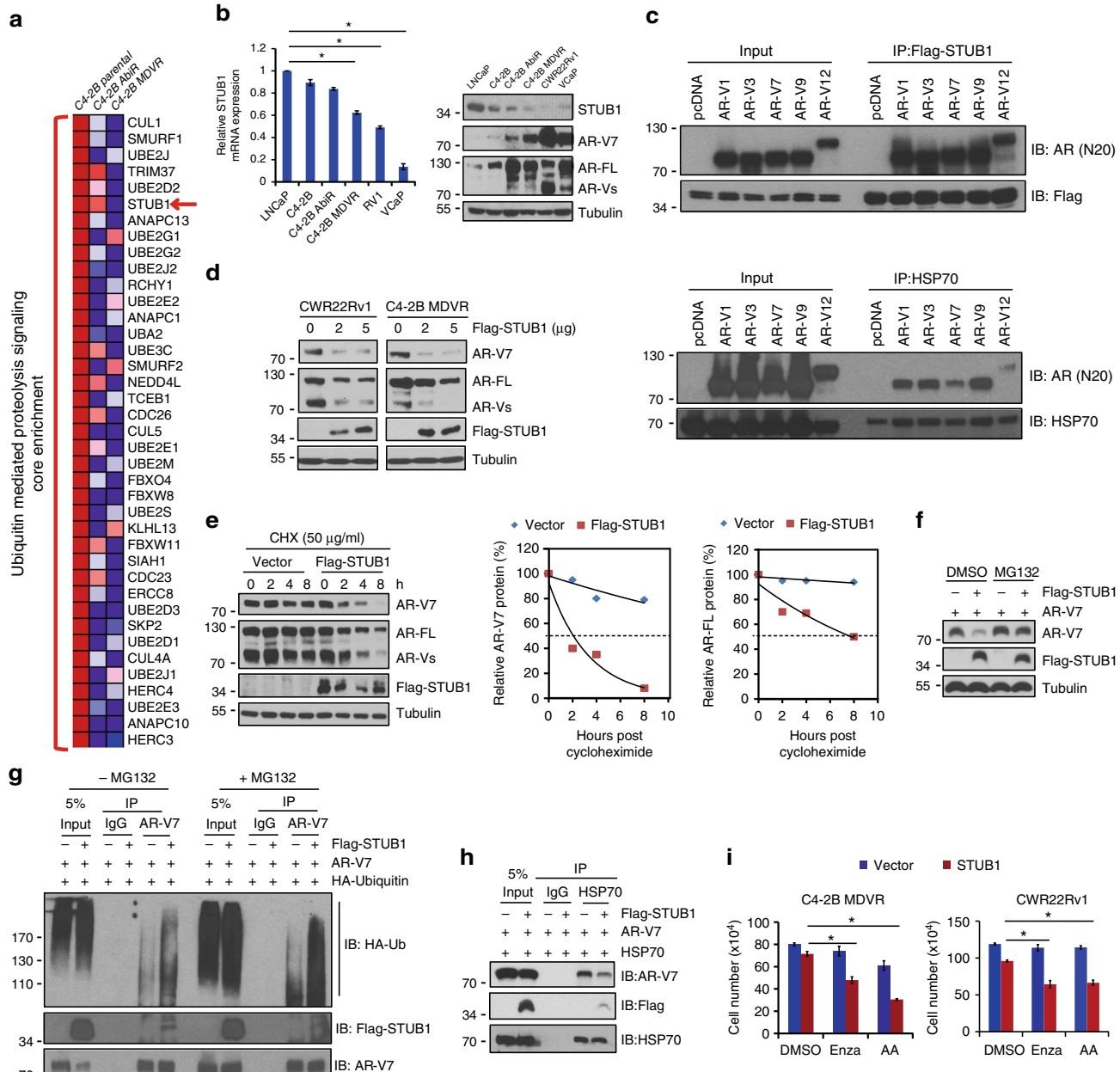

**Fig. 2** STUB1/HSP70 complex regulates AR variants protein expression. **a** Core-enrichment genes of ubiquitin-mediated proteolysis pathway in C4-2B parental, C4-2B MDVR, and C4-2B AbiR cells. The red arrow showed STUB1 was suppressed in resistant cells. **b** mRNA levels of STUB1 in LNCaP, C4-2B parental, C4-2B AbiR, C4-2B MDVR, CWR22Rv1, and VCaP cells was examined by real-time PCR (left). Protein expression of STUB1, AR-V7, AR-FL was determined by western blot (right). **c** 293 cells were transiently transfected with Flag-STUB1 or HSP70 with or without vector, AR-V1, AR-V3, AR-V7, AR-V9, or AR-V12 for 3 days and whole cell lysates were immunoprecipitated with anti-Flag or HSP70 antibodies and bloted with indicated antibodies. **d** CWR22Rv1 and C4-2B MDVR cells were transiently transfected with 2 or 5 μg Flag-STUB1 for 3 days, whole cell lysates were collected and subjected to western blot. **e** CWR22Rv1 cells were transiently transfected with vector or flag-STUB1 and then treated with 50 μg/mL cycloheximide, total cell lysates were collected at 0, 2, 4, and 8 h after the treatment and subjected to western blot. Half-lives of AR-V7 and AR-FL were calculated. **f** 293 cells were transiently transfected with AR-V7 with or without STUB1 for 3 days and then treated with 5 μM MG132 overnight. Total cell lysates were collected and subjected to western blot. **g** 293 cells were transiently transfected with AR-V7, flag-STUB1, and HA-ubiquitin plasmids for 3 days, following treatment with or without 5 μM MG132 for additional 6 h. Total cell lysates were collected and immunoprecipitated with AR-V7 antibody and blotted with indicated antibodies. **h** 293 cells were co-transfected with HSP70, AR-V7 with or without Flag-STUB1 for 3 days and then whole cell lysates were immunoprecipitated with anti-HSP70 antibody and blotted with indicated antibodies. **i** C4-2B MDVR or CWR22Rv1 cells were transiently transfected with STUB1 and then treated with 20 μM enzalutamide or 5 μM abiraterone for 3 days and total cell number was determined. *$p < 0.05$. Results are the mean of three independent experiments (±s.d.). Statistical analysis was performed using one-way ANOVA. AR-FL: full-length AR, AR-Vs: AR-Variants, Ub: Ubiquitin

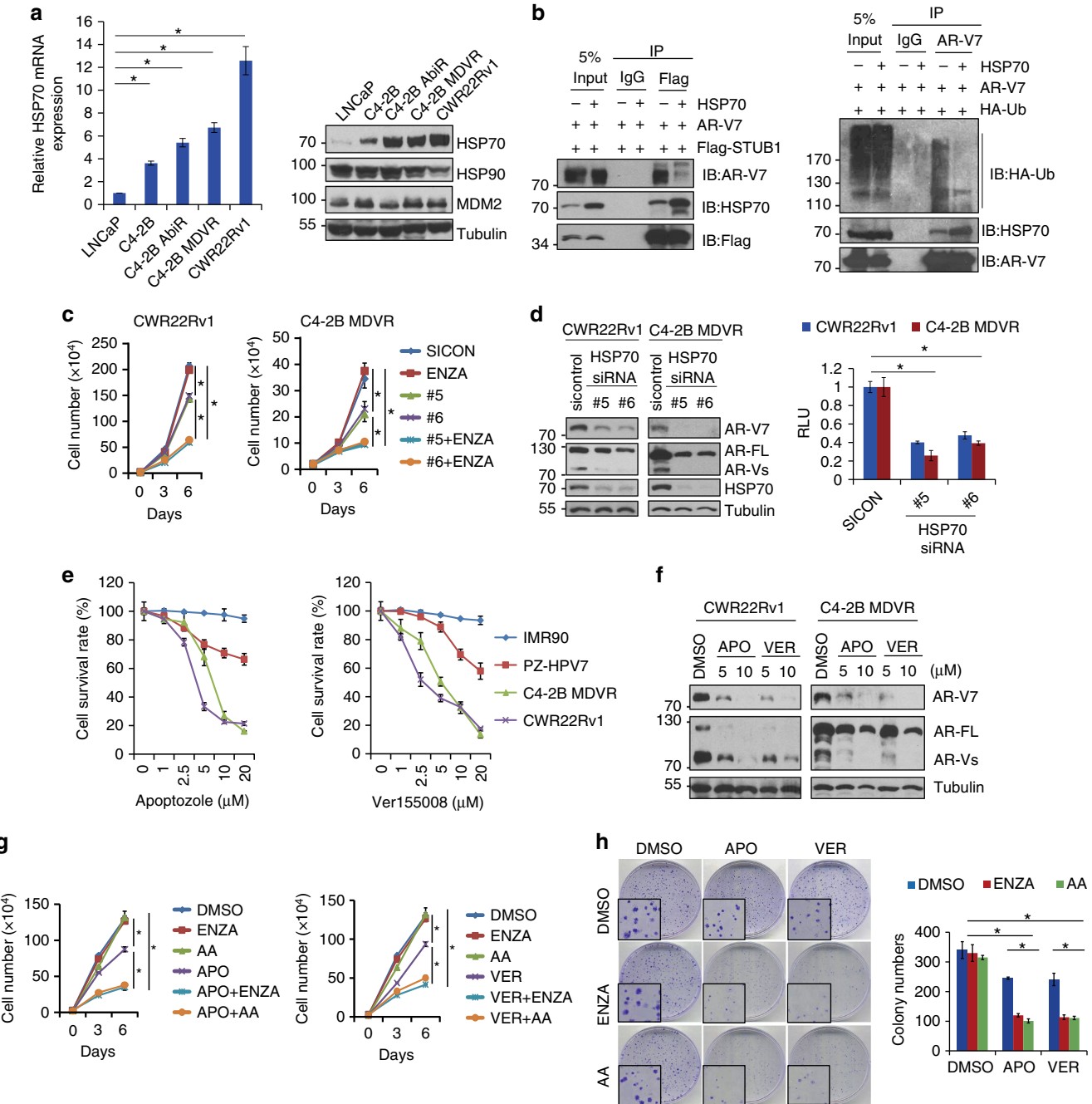

**Fig. 3** STUB1/HSP70 complex regulates sensitivity to anti-androgens. **a** Total mRNA and whole cell lysates from LNCaP, C4-2B, C4-2B AbiR, C4-2B MDVR, and CWR22Rv1 cells were collected and subjected to qRT-PCR and western blot, respectively. **b** 293 cells were co-transfected with Flag-STUB1, AR-V7, HA-Ub with or without HSP70 for 3 days, whole cell lysates were immunopreciptated with anti-Flag or AR-V7 antibodies and blotted with AR-V7, HSP70, Flag or HA antibodies. **c** CWR22Rv1 and C4-2B MDVR cells were transiently transfected with two independent HSP70 siRNA (#5 and #6) and then treated with 20 μM enzalutamide. Total cell number was determined at 3 and 6 days. **d** C4-2B MDVR and CWR22Rv1 cells were transiently transfected with two independent HSP70 siRNA, whole cell lysates were collected and subjected to western blot. PSA luciferase activity was determined. **e** IMR90, PZ-HPV7, C4-2B MDVR, and CWR22Rv1 cells were treated with different concentrations of APO and VER for 3 days, total cell numbers were determined. **f** C4-2B MDVR and CWR22Rv1 cells were treated with 5 and 10 μM APO or VER for 48 h, total cell lysates were collected and subjected to western blot. **g** C4-2B MDVR cells were treated with 5 μM APO and VER with or without 20 μM enzalutamide or 5 μM abiraterone, total cell number was determined at 3 and 6 days. **h** C4-2B MDVR cells were treated with 5 μM APO or VER with or without enzalutamide and abiraterone, colonogenic assay was performed and colonies were quantified. *$p < 0.05$. Results are the mean of three independent experiments (±s.d.). Statistical analysis was performed using one-way ANOVA. AR-FL: full-length AR, AR-Vs: AR-Variants, Ub: Ubiquitin, ENZA: Enzalutamide, AA: Abiraterone acetate, APO: Apoptozole, VER: Ver155008, RLU: Relative luciferase unit

Next, we examined the combinational effects of APO or VER with next generation anti-androgens. As shown in Fig. 3g and Supplementary Fig. 5a, both APO and VER profoundly enhanced enzalutamide and abiraterone treatments in CWR22Rv1 and C4-2B MDVR cells in a time-dependent and dose-dependent manner. However, there were no combination effects in PZ-HPV7 and IMR90 cells (Supplementary Fig. 5b). These results were also confirmed by clonogenic assay (Fig. 3h and Supplementary Fig. 5c). To further demonstrate these two HSP70 inhibitors in clinical applications, conditionally reprogrammed cells (CRCs) were established from a Gleason 10 score prostate cancer patient based on the published protocol [30]. As shown in Supplementary Fig. 6a, these cells displayed highly heterogeneous phenotype and overexpressed AR and HSP70 in both cytoplasm and nucleus. These cells were resistant to enzalutamide and abiraterone treatment but showed dose response to APO and VER treatment (Supplementary Fig. 6b). Combination of APO or VER with enzalutamide further suppressed cell proliferation in these CRCs (Supplementary Fig. 6c). Taken together, these results suggest that targeting HSP70 can improve next generation anti-androgen treatment in advanced prostate cancer.

**HSP70 inhibitors promote AR-V7 ubiquitination via STUB1.** To determine if inhibition of HSP70 promotes AR-V7 degradation through enhanced ubiquitination, 293 cells overexpressed with AR-V7, HSP70, and HA-Ubiquitin were treated with either 20 μM VER or 20 μM APO overnight, along with 5 μM MG132 during the incubation to prevent degradation of ubiquitinated AR-V7. We found that both APO and VER resulted in enhanced ubiquitination of AR-V7 compared to DMSO-treated control cells (Fig. 4a top). The results were also observed in C4-2B MDVR cells (Fig. 4a bottom). Additionally, both APO and VER significantly increased the binding of AR-V7 to STUB1 in 293 cells (Fig. 4b left) and AR/AR variants to STUB1 in C4-2B MDVR cells (Fig. 4b right), suggesting that binding of HSP70 to AR-V7 potentially protects AR-V7 protein from degradation, and inhibition of HSP70 promotes STUB1 and AR-V7 binding which leads to AR-V7 protein degradation. Additionally, we confirmed our findings by dual immunofluorescence staining. As shown in Fig. 4c, STUB1 and AR-V7 were not co-localized in 293 cells when HSP70 was overexpressed. AR-V7 was dominantly present in the nucleus; however, STUB1 was mostly localized in the cytoplasm when HSP70 was overexpressed. APO and VER treatment significantly enhanced AR-V7 and STUB1 co-localization. We also found that knockdown of STUB1 in C4-2B MDVR cells rescued AR and AR-V7 suppression by APO and VER treatment, suggesting that AR/AR-V7 degradation induced by APO and VER is mediated by STUB1 (Fig. 4d). Furthermore, AR-V7 and HSP70 overexpression significantly rescued APO and VER-mediated growth inhibition in prostate cancer cells. As shown in Fig. 4e, overexpresstion of both HSP70 and AR-V7 into C4-2B cells significantly increased cell growth in CS-FBS condition. APO and VER still suppressed the cell proliferation but showed higher rate than the vector transfected and single HSP70 or AR-V7-transfected cells. HSP70 overexpression stabilized AR/AR-V7 protein in C4-2B cells (Fig. 4f). These results suggest that HSP70 stabilizes and protects AR-V7 from degradation. Inhibition of HSP70 promotes AR-V7 ubiquitination and degradation by STUB1.

**HSP70 inhibition suppresses AR-FL and AR-V7 signaling.** To further explore the gene regulating mechanisms underlying the downregulation of HSP70 in drug-resistant prostate cancer cells, we performed RNA sequencing analyses using C4-2B MDVR and CWR22Rv1 cells treated with APO or VER to identify gene programs that are affected by HSP70 inhibition. There are 10,773 genes and 10,317 genes that were differentially expressed in APO or VER treated C4-2B MDVR cells, respectively, and 7905 genes that were commonly regulated by both APO and VER (fold change > 1.2; Fig. 5a). The top pathways upregulated by HSP70 inhibition include unfolded protein response (UPR), the p53 pathway and post translational protein modification pathway. The down-regulated pathways include cell cycle, androgen response, E2F targets, and Myc targets as analyzed by GSEA (Fig. 5b and Supplementary Tables 2–5). APO and VER-regulated genes were mainly clustered in two major groups compared with DMSO treatment as plotted by heatmap using hierarchical clustering with the genes found as commonly regulated by APO and VER, indicating a high degree of concordance in the expression changes that were induced by APO and VER treatment (Fig. 5c left). At the individual gene level, we observed upregulation of UPR genes (for example, XIAP1, ATF4, and ATF6) in both APO and VER-treated cells. We also found that AR and AR-V7-regulated genes (for example, KLK3, FKBP5, UBE2C, and AKT1) were suppressed by APO and VER treatment (Fig. 5c right). Further GSEA revealed AR and AR-V7 pathways were significantly blocked by APO and VER treatment in both drug-resistant cell lines. As shown in Fig. 5d left, the classical PID-AR pathway was significantly suppressed. Both APO and VER robustly disrupted AR and AR-V7 target gene programs (Fig. 5d right). qRT-PCR verified that AR and AR-V7 target genes, such as KLK2, KLK3, NKX3-1, FKBP5, UBE2C, and Myc were suppressed by both APO and VER treatment. Notably, genes such as UBE2C and Myc, which are preferentially upregulated by AR-V7, were significantly suppressed by APO and VER treatment (Fig. 5e and Supplementary Fig. 7).

**HSP70 inhibition enhances enzalutamide therapy in vivo.** To examine if targeting HSP70 enhances enzalutamide treatment in vivo, we generated enzalutamide-resistant xenografts derived from CWR22Rv1 cells, as well as HSP70 and AR-V7 over-expressing LuCaP35CR PDX model and treated with APO and VER (Fig. 6a). CWR22Rv1 tumors were resistant to enzalutamide treatment ($p = 0.73$), both APO and VER significantly inhibited tumor growth ($p = 0.0095$ and $p = 0.016$, respectively). Combination of APO or VER with enzalutamide further inhibited tumor growth of CWR22Rv1 xenografts ($p = 0.0012$ and $p = 0.0045$, respectively) (Fig. 6b, c). Survival was improved in the APO and VER groups compared to the vehicle or enzalutamide-treated groups. Combination treatment of APO or VER with enzalutamide further improved animal survival (Fig. 6d). Treatments did not affect mouse body weights (Supplementary Fig. 8a, left). Vital organs, such as liver and kidney, were harvested for histopathologic examination and no significant pathological changes were found in the organs from any group. As shown in Supplementary Fig. 8b, livers did not show any vacuolar changes; and there was no sign of inflammation at the renal pelvis in single or combination treatment groups. Immunohistochemical staining of AR-V7 and Ki67 showed AR-V7 expression and cell proliferation were significantly inhibited by APO or VER treatment alone and further inhibited by the combination treatments (Fig. 6e).

To further characterize the effects of HSP70 inhibition on tumor growth in vivo, the LuCaP 35CR PDX model was utilized as described in Fig. 6f. As shown in Fig. 6g, h, enzalutamide slightly suppressed LuCaP 35CR tumor growth but did not reach significance ($p = 0.54$), APO significantly suppressed LuCaP 35CR growth ($p = 0.026$), while combination of APO and enzalutamide further reduced tumor growth ($p = 0.0087$). Treatments did not alter mouse body weights (Supplementary Fig. 8a right). Enzalutamide treatment slightly, but insignificantly,

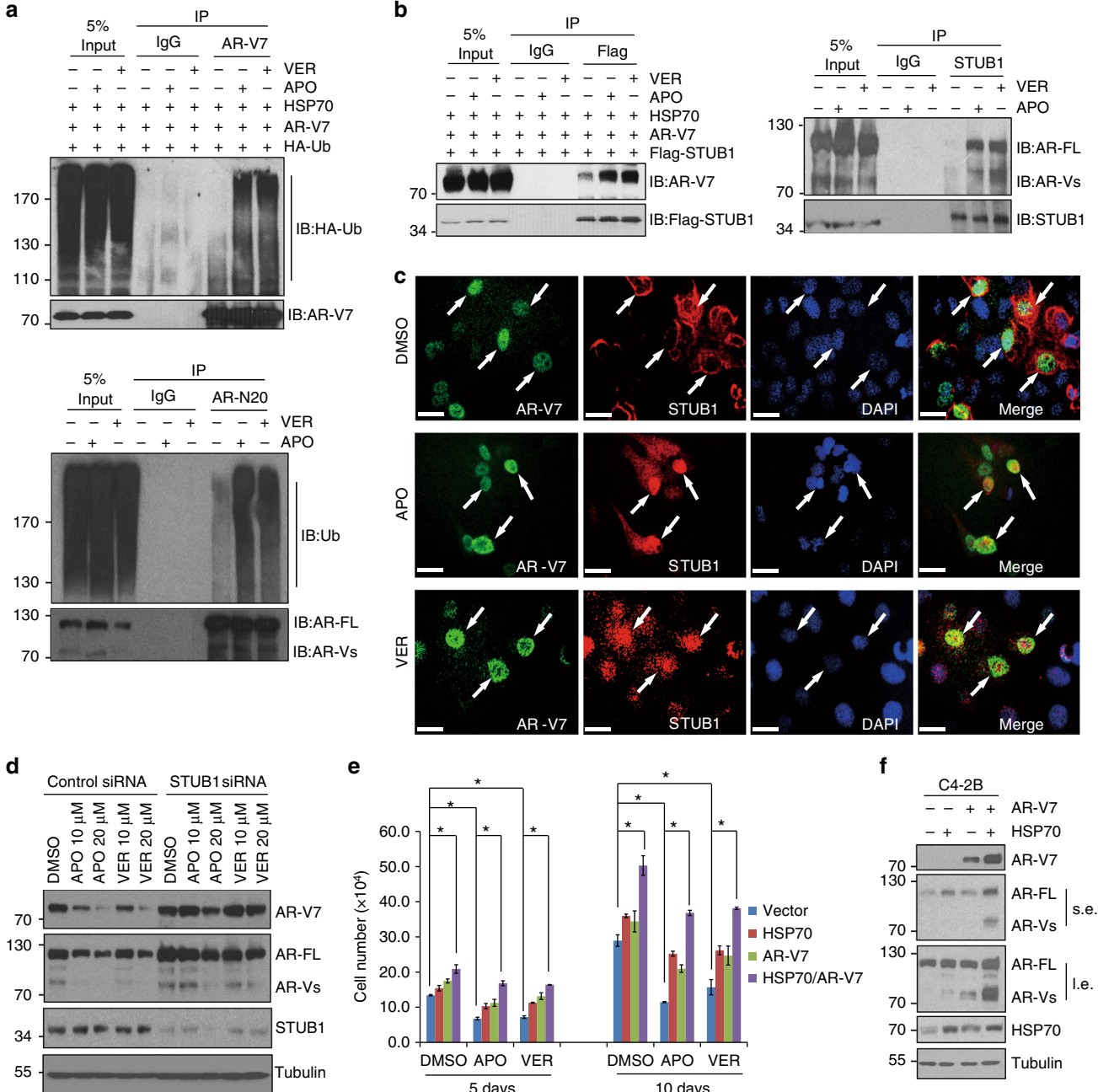

**Fig. 4** HSP70 inhibitors promote AR-V7 ubiquitination via STUB1. **a** Immunoblots following anti-AR-V7 immunoprecipitation from AR-V7, HSP70, and HA-Ubiquitin overexpressed 293 cells that were co-treated overnight with 20 μM APO or VER with 5 μM MG132 to inhibit degradation of ubiquitinated AR-V7 (top). C4-2B MDVR cells were treated with 20 μM APO or VER overnight with 5 μM MG132 for 6 h, total cell lysates were immunoprecipitated with anti-AR (N20) antibody and immunoblotted with anti-ubiquitin and AR-FL antibodies (bottom). **b** AR-V7, HSP70, and Flag-STUB1 were overexpressed in 293 cells and then treated with APO or VER overnight. Whole cell lysates were subjected to anti-Flag IP, AR-V7, and Flag were blot (left). C4-2B MDVR cells were treated with 20 μM APO or VER overnight with 5 μM MG132 for 6 h, total cell lysates were immunoprecipitated with anti-STUB1 antibody and immunoblotted with anti-AR, STUB1 antibodies (right). **c** 293 cells were co-transfected with AR-V7, HSP70, and Flag-STUB1 for 3 days, following treatment with APO or VER for 24 h, AR-V7 and STUB1 were visualized by dual immunofluorescence staining. White arrows indicated the typical staining cells in each group. Scale bar 20 μm. **d** C4-2B MDVR cells were transiently transfected with STUB1 siRNA for 3 days, following treatment with APO or VER overnight; whole cell lysates were collected and subjected to western blot. **e, f** C4-2B cells were transiently transfected with vector, AR-V7, HSP70, or AR-V7 plus HSP70, following treatment with APO or VER, the cell growth was determined at 5 and 10 days. The whole cell lysates were collected and subjected to western blot. *$p < 0.05$. Results are the mean of three independent experiments (±s.d.). Statistical analysis was performed using one-way ANOVA. AR-FL: full-length AR, AR-Vs: AR-Variants, Ub: Ubiquitin, APO: Apoptozole, VER: Ver155008, s.e.: short exposure, l.e.: long exposure

suppressed tumor PSA expression ($p = 0.28$), APO treatment significantly suppressed PSA ($p = 0.029$), and the combination treatment further reduced the PSA levels ($p = 0.0053$) (Fig. 6i). APO and the combination treatments also improved survival compared to either vehicle or enzalutamide treatment alone (Fig. 6j). Immunohistochemical staining of Ki67 showed cell proliferation was significantly inhibited by APO, and further inhibited by the combination treatment (Fig. 6k). Taken together,

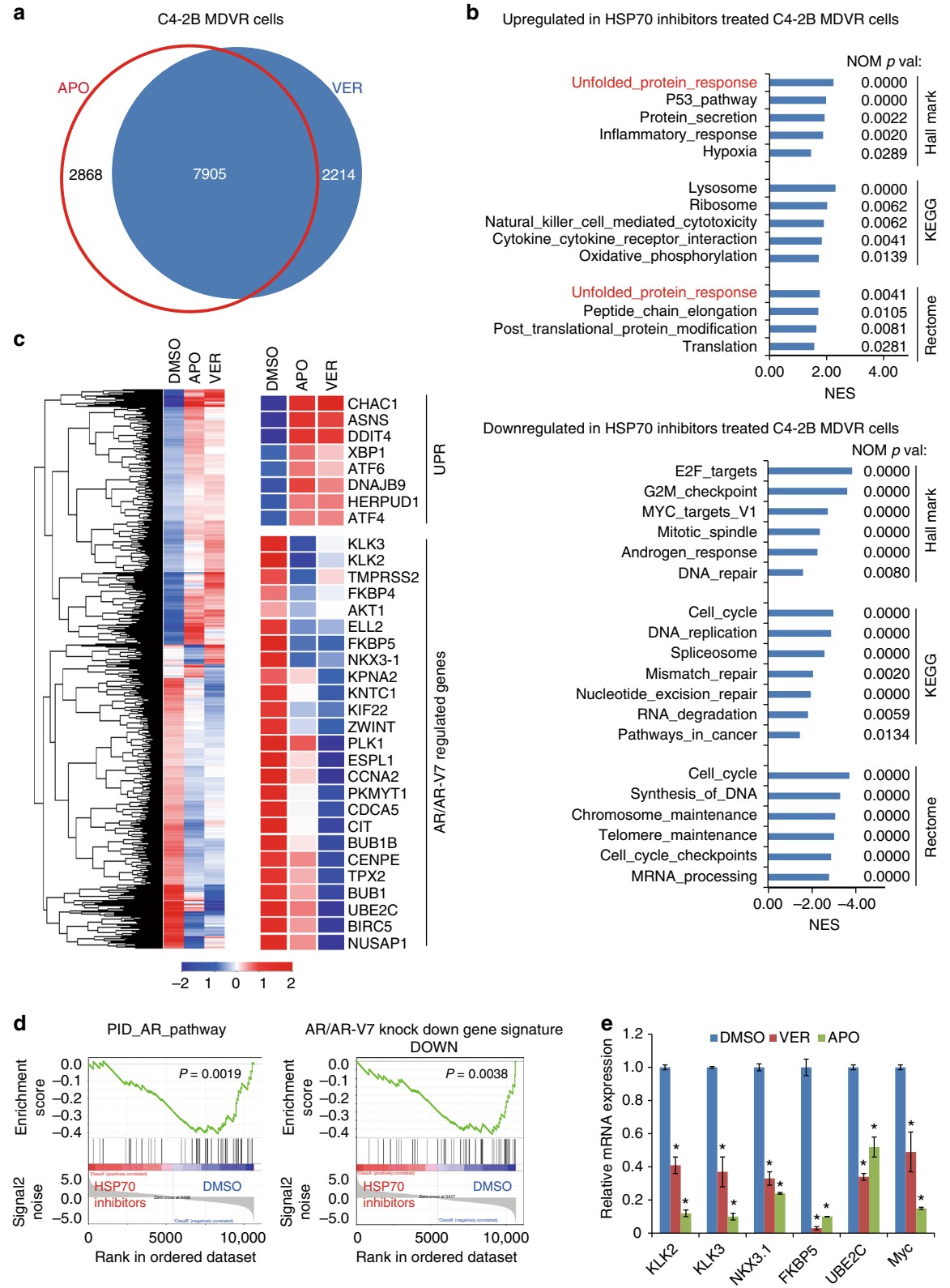

these results suggest that inhibition of HSP70 by APO or VER reduces enzalutamide-resistant tumor growth, and combination of enzalutamide with either APO or VER further suppresses tumor growth.

**HSP70 level is correlated with AR/AR-V7 in prostate tumors**. To determine the relationship between HSP70 and AR expression

in human prostate cancer, we first analyzed HSP70 and AR-FL expression in GEO and Oncomine databases. Levels of HSP70 and AR-FL are significantly upregulated in metastatic castration-resistant prostate cancer (mCRPC) patients compared to benign prostate and primary prostate cancer in four independent GEO databases (Fig. 7a). In addition, the level of HSP70 expression is significantly increased in high Gleason score prostate cancer in

**Fig. 5** HSP70 inhibitors suppress AR and AR-V7 signaling. **a** Venn diagram of RNA-seq analysis of the two comparisons: APO vs. DMSO and VER vs. DMSO in C4-2B MDVR cells. **b** GSEA of top enriched gene sets in C4-2B MDVR cells treated by HSP70 inhibitors. The upregulated and down regulated gene sets from the Hallmark, KEGG and Rectome platforms were output by GSEA. **c** Heatmap and hierarchical clustering of the differentially expressed genes (DEGs) between APO and VER treatment in C4-2B MDVR cells with fold change (FC) > 1.2, as compared to vehicle (DMSO). The genes were displayed in rows and the normalized counts per sample were displayed in columns. Red indicates up-regulated and blue designates down-regulated expression levels. Middle and right, UPR, AR, and AR-V7 activity-signature genes that were altered in expression are displayed. **d** GSEA of the PID-AR pathway in C4-2B MDVR cells treated with HSP70 inhibitors, as compared to DMSO (left). GSEA of the AR and AR-V7 gene signatures in C4-2B MDVR cells treated with HSP70 inhibitors (right). The signature was defined by genes that underwent significant expression changes as a result of AR and AR-V7 knockdown in prostate cancer cells 6. **e** qRT-PCR analysis of the indicated genes in C4-2B MDVR cells treated with DMSO or with HSP70 inhibitors (10 μM) for 48 h. $*p < 0.05$. Results are the mean of three independent experiments (±s.d.). Statistical analysis was performed using two tailed Student's $t$-test. APO apoptozole, VER Ver155008

two independent datasets from the Oncomine database (Fig. 7b). To examine whether HSP70 expression is correlated with AR and AR-V7 expression in advanced prostate cancer, 26 high Gleason score (≥8) patients' samples were collected and the levels of HSP70, AR-V7, along with HSP90 and AR-FL were examined by qRT-PCR (Supplementary Table 1). As shown in Fig. 7c, d, HSP70 levels were significantly correlated with AR-V7 levels ($r = 0.792$, $p = 0.00000143$) and AR-FL levels ($r = 0.689$, $p = 0.00000993$). However, there is no significant correlation between HSP90 and AR-V7/AR-FL in these samples (Supplementary Fig. 9a). Figure 6e shows the expression of HSP70 protein in a representatively low (Gleason 6) and high (Gleason 9) Gleason grade prostate cancer by IHC. We also interrogated the GSE32269 and GSE6919 databases including 51 and 90 prostate tumor samples, respectively. As shown in Supplementary Fig. 9b, HSP70 significantly correlated with AR-FL in both databases. Collectively, these results suggest that HSP70 is significantly overexpressed in advanced stage prostate cancer and the level of HSP70 expression is correlated with AR-V7 and AR-FL expressions.

## Discussion

Our study uncovers a balanced crosstalk between proteostasis and next generation anti-androgen resistance through regulation of the STUB1/HSP70/AR-V7 complex. We discovered that the ubiquitin-mediated proteolysis pathway and proteasome activity are suppressed in enzalutamide and abiraterone-resistant prostate cancer cells, and play a critical role in the degradation of the AR and its variants, particularly AR-V7. Modulation of AR and AR-V7 proteostasis balance could be achieved by inhibition of HSP70, which may provide a valuable strategy to overcome resistance to next generation anti-androgen therapies. Our findings suggest that alteration of the chaperone–ubiquitin–proteasome system may represent a general mechanism for the regulation of AR variants protein stability. With that, we provide rationale targeting proteostasis through inhibition of HSP70 as a potential therapeutic strategy to overcome drug resistance to AR-targeted therapies in CRPC.

Emerging evidence suggests that proteomic instability, such as protein misfolding and aggregation play pivotal roles in cancer cell survival and progression[31]. Proteomic equilibrium may be altered during tumorigenesis, and consequently leads to oncogenic activation at the protein level[32]. Our study determined that the imbalance of proteostasis brought on by anti-androgen treatment in prostate cancer cells might be a critical mechanism conferring drug resistance. Deficiency of proteostasis and lack of proteasome activity in enzalutamide and abiraterone-resistant prostate cancer might trigger overexpression of onco-proteins, such as AR and its variants through an inability of protein clearance. Previous reports suggested that constitutively active AR variants confer the CRPC phenotype and resistance to next generation anti-androgens in both pre-clinical and clinical

models[6,7,33–35]. Among these AR variants, overexpression of AR-V7 and AR-V9 was reported in enzalutamide and abiraterone-resistant prostate cancer patients, which is consistent with our findings in this study[7,36]. The literature suggests that AR variants are generated from intragenic AR gene rearrangements that alter AR gene structure[11] and aberrant mRNA splicing and premature translation termination evident in prostate cancer cell lines and xenografts[10]. In the present study, we determined that AR variants are not only generated through mRNA splicing but also through protein stabilization via protein ubiquitin proteasome alteration in drug-resistant prostate cancer. The half-life of AR-V7 is significantly extended in enzalutamide-resistant prostate cancer cells compared with parental cells, suggesting that next generation anti-androgen treatments might alter the prostate cancer ubiquitin-proteolysis system and stabilize AR-V7 protein. Notably, the proteasome activity is significantly suppressed in enzalutamide and abiraterone-resistant prostate cancer cells and the E3 ligase STUB1 and its binding chaperone protein HSP70 might control the AR-V7 proteostasis and confer the resistance.

STUB1 is one of the important E3 ligases regulating several nuclear receptors, such as GR[37], ERα[38], and AR[39–41]. Overexpression of STUB1 in cultured cells promotes ubiquitination of cystic fibrosis transmembrane conductance regulator (CFTR), c-Myc and Raf kinase[42–45]. The importance of STUB1 involved in oncogenic proteins regulation was reinforced by our discovery that STUB1 bound with AR/AR-V7 and enhanced their ubiquitination and degradation. The AR-V7 co-chaperone protein, HSP70, formed a complex with AR-V7 and assisted AR-V7 protein maturation. STUB1 blocked HSP70 and AR/AR-V7 complex formation, leading to AR/AR-V7 protein degradation. Targeting HSP70 enhances STUB1 and AR/AR-V7 binding and leads to AR/AR-V7 ubiquitination and degradation (Fig. 7f). Our findings related to the HSP70/STUB1/AR-V7 complex are important as this mechanism may represent a general chaperone–ubiquitin–proteasome mechanism for the regulation of AR variants protein stability that may involve in treatment resistance in advanced prostate cancer.

HSP70 consists of an N-terminal ATPase domain (or nucleotide-binding domain) and a C-terminal substrate-binding domain (SBD) that recognizes polypeptide substrates. HSP70s together with HSP40s are the prominent chaperone families involved in chaperone-assisted proteasomal degradation of misfolded proteins[46–48]. After misfolded protein is depleted in the cells, STUB1 mediates the HSP70 turnover itself and reduces HSP70 to physiological level 19. Here we discovered that HSP70 was overexpressed in anti-androgen-resistant prostate cancer cells and mCRPC tumors, which is consistent with reports showing that HSP70 is overexpressed in various cancers[49–51]. Notably, HSP70 expression was correlated with AR-V7 level in high Gleason score prostate tumors. The homeostasis of the HSP70/STUB1 complex might play a substantial role in controlling AR-V7 protein levels. HSP70 binds with AR-V7 and protects it from

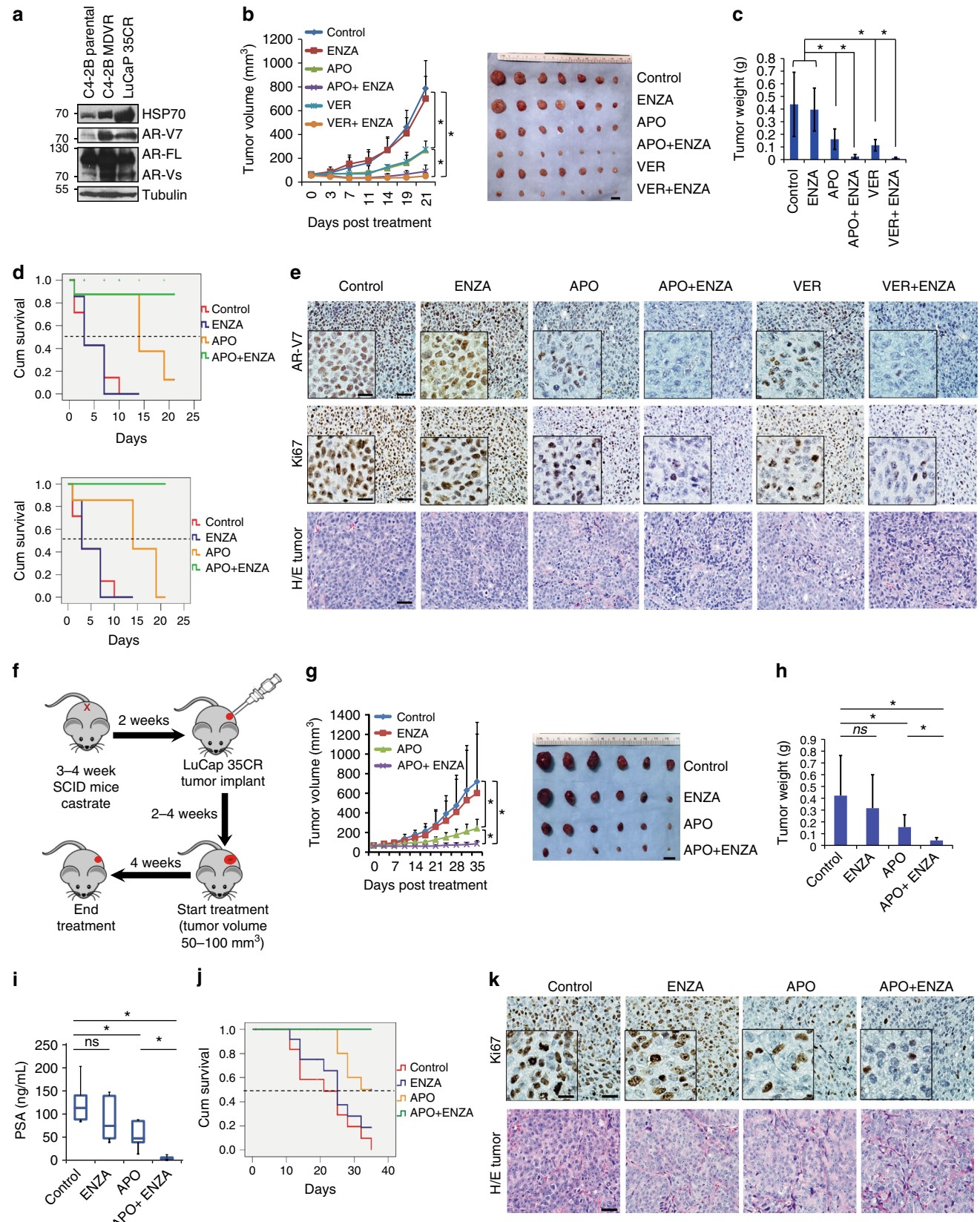

degradation by STUB1. Our study suggests that STUB1 remains in the cytoplasm while HSP70 is overexpressed. HSP70 inhibition significantly promotes STUB1 entering the nucleus and binds to AR-V7. Therefore, targeting HSP70 might be a logical approach to treat AR-V7 overexpressing CRPC patients. Using specific siRNA to target HSP70 or HSP70 inhibitors to suppress HSP70 activity significantly blocked AR-V7 expression and suppressed its transcriptional activity. These results suggest that HSP70 is a

**Fig. 6** HSP70 inhibitors enhance enzalutamide treatment in vivo. **a** Total cell lysates from C4-2B parental, C4-2B MDVR cells, and LuCaP35CR xenograft tumors were extracted and subjected to western blot. HSP70, AR-V7, and AR-FL protein expression levels were determined. **b, c** Mice bearing CWR22Rv1 xenografts were treated with vehicle control, enzalutamide (25 mg/kg p.o.), APO (5 mg/kg i.p.), VER (15 mg/kg i.p.), APO plus enzalutamide, or VER plus enzalutamide for 3 weeks ($n = 7$). Tumor volumes were measured twice weekly. Tumors were photographed and weighed. Scale bar 1 cm. Data represent means ± s.d. from 7 mice per group. **d** Kaplan–Meier curves showing survival benefits of HSP70 inhibitors single treatment, HSP70 inhibitors, and enzalutamide combination treatment in CWR22Rv1 xenograft tumors. **e** IHC staining of AR-V7 and Ki67 in each group was performed. Scale bar 50 μm (outside) and 20 μm (inside). **f** Treatment scheme on LuCaP 35CR PDX model. **g, h** Mice bearing LuCaP 35CR xenografts were treated with vehicle control, enzalutamide (25 mg/kg p.o.), APO (5 mg/kg i.p.), or their combination for 5 weeks ($n = 6$). Tumor volumes were measured twice weekly. Tumors were photographed and weighed. Scale bar 1 cm. Data represent means ± s.d. from six mice per group. **i** PSA expression in mice serum was examined in different treatment groups. **j** Kaplan–Meier curves showing survival benefits of APO single treatment, APO and enzalutamide combination treatment in LuCaP 35CR tumors. **k** IHC staining of Ki67 in each group was performed. Scale bar 50 μm (outside) and 20 μm (inside). *$p < 0.05$. Statistical analysis was performed using one-way ANOVA. AR-FL: full-length AR, AR-Vs: AR-Variants, ENZA: Enzalutamide, APO: Apoptozole, VER: Ver155008

potential therapeutic target that drives AR-V7 expression, drug resistance, and prostate cancer progression.

Studies over the last decades support the concept of targeting chaperone proteins in cancer therapy. The most well-known therapies targeting chaperone proteins currently in drug development are HSP90 inhibitors. HSP90 assists several oncogenic proteins including AR and facilitates their maturation[52]. However, HSP90 binds with the AR ligand-binding domain (LBD) which is missing in constitutively active AR variants. This makes HSP90 inhibition irrelevant in the treatment of AR variant dominated prostate cancer. Additionally, targeting HSP90 significantly increased HSP70 expression, suggesting a compensating role of HSP70 as co-chaperone protein[53]. Several categories of HSP70 inhibitors have been pursued in drug development, such as targeting the peptide-binding domain (PBD), amino-terminal ATPase domain (ABD), or HSP70 co-chaperones[54]. In the present study, we provide a proof of concept by using HSP70 inhibitors (APO and VER) targeting ABD to treat AR-V7 overexpressing and drug-resistant prostate cancer. Through the bioinformatics analysis, we found that inhibition of HSP70 using APO and VER suppresses AR/AR-V7 signaling pathways in resistant cells. Additionally, APO and VER activate the UPR pathway and suppress cell cycle. UPR activation induces the caspases which localize to the endoplasmic reticulum (ER) membrane and trigger apoptotic pathways[55]. Notably, tumors that are resistant to enzalutamide are highly sensitive to APO and VER treatments. APO and VER significantly enhanced enzalutamide treatment both in vitro and in vivo. Our promising pre-clinical data shed light on future clinical trial development by using HSP70 inhibitors in advanced prostate cancer treatment.

In conclusion, our study suggests that the ubiquitin proteasome system is suppressed in enzalutamide and abiraterone-resistant prostate cancer models. The STUB1/HSP70 complex is involved in AR and AR variant stabilization and regulates the sensitivity to next generation AR-targeted therapy. Additionally, we provide a proof of concept study showing that targeting HSP70 could be a valuable strategy to treat AR-V7 overexpressing CRPC and improve enzalutamide treatment. Clinically, HSP70 level is correlated with AR and AR-V7 in high Gleason score prostate tumors, suggesting that HSP70 might serve as a potential marker to indicate prostate cancer progression and therapeutic resistance.

## Methods

**Cells lines and tissue culture**. LNCaP, C4-2B, and CWR22Rv1 were maintained in RPMI1640 supplemented with 10% fetal bovine serum (FBS), 100 units per ml penicillin, and 0.1 mg per ml streptomycin. IMR90, 293, and VCaP cells were maintained in DMEM supplemented with 10% FBS, 100 units per ml penicillin, and 0.1 mg per ml streptomycin. PZ-HPV7 cells were maintained in keratinocyte serum-free medium (K-SFM) with the required supplements (Invitrogen). All experiments with cell lines were performed within 6 months of receipt from the American Type Culture Collection (ATCC) or resuscitation after cryopreservation. C4-2B cells were kindly provided and authenticated by Dr. Leland Chung, Cedars-

Sinai Medical Center (Los Angeles, CA). The resistant cells were isolated and referred to as C4-2B MDVR (C4-2B enzalutamide resistant) and C4-2B AbiR (C4-2B abiraterone resistant)[56,57]. C4-2B MDVR and C4-2B AbiR were maintained in 20 μM enzalutamide containing medium and 10 μM abiraterone acetate containing medium, respectively. Parental C4-2B cells were passaged alongside the resistant cells as an appropriate control. All cell lines have been routinely tested mycoplasma free by PCR and authenticated by short tandem repeat (STR) method. All cells were maintained at 37 °C in a humidified incubator with 5% carbon dioxide. Enzalutamide, abiraterone acetate, Apoptozole (APO), and Ver155008 (VER) were purchased from Selleck Chemicals.

**Plasmids and cell transfection**. For small interfering RNA (siRNA) transfection, cells were seeded at a density of $0.5 \times 10^5$ cells per well in 12-well plates or $2 \times 10^5$ cells per well in six-well plates and transfected with 20 nM of siRNA (Invitrogen) targeting the HSP70 sequence (HSPA1A/HSPA1B, Catalog# 262305 and 262306), STUB1 sequence (Catalog# 215046), or control siRNA (Catalog# 12935300) using Lipofectamine-iMAX (Invitrogen). The effect of siRNA-mediated gene silencing was examined using qRT-PCR and western blot 2–3 days after transfection. Cells were transiently transfected by expressing plasmids for vectors, AR-FL, AR-V1, AR-V3, AR-V7, AR-V9, AR-V12 (AR-V567es), Flag-STUB1, HA-Ubiquitin, or HSP70 (HSPA1B, OriGene, Catalog# SC116767) using Lipofectamine 2000 (Invitrogen).

**Protein extraction and western blotting**. Whole cell protein extracts were resolved on SDS–PAGE and proteins were transferred to nitrocellulose membranes. After blocking for 1 h at room temperature in 5% milk in PBS/0.1% Tween-20, membranes were incubated overnight at 4 °C with the indicated primary antibodies AR (441), AR (N-20), AR (C-19), HSP70 (F-3 and H-300), STUB1 (H231 and G-2), MDM2 (HDM2-323), HSP90 (4F10), HA (F-3), Ubiquitin (P4D1 and FL76), 1:1000 dilution, Santa Cruz Biotechnology, Santa Cruz, CA; STUB1 (C3B6, 1:100 for IP, Cell Signaling antibody); AR-V7 (AG10008, Mouse monoclonal antibody, 1:1000 dilution, precision antibody); FLAG® M2 monoclonal antibody (F1804, 1:1000 dilution for western blot, 1:200 for IP, Sigma-Aldrich, St. Louis, MO); and Tubulin (T5168, monoclonal anti-α-tubulin antibody, 1:5000 dilution, Sigma-Aldrich, St. Louis, MO). Tubulin was used as loading control. Following secondary antibody incubation, immunoreactive proteins were visualized with an enhanced chemiluminescence detection system (Millipore, Billerica, MA). All uncropped scans of Western Blot are attached in Supplementary Figs. 10 and 11.

**Luciferase reporter assay**. C4-2B, CWR22Rv1, or C4-2B MDVR cells were plated into 12-well plates ($1 \times 10^5$) and grown to 80% confluence and transiently transfected using Lipofectamine 2000 (Invitrogen). pGL3-PSA6.0 luciferase construct was co-transfected with pRL-TK (TK promoter-Renilla luciferase construct as internal control). Briefly, pGL3-PSA6.0 luciferase construct and pRL-TK along with HSP70 siRNA, AR-V7, or HSP70 were mixed and transfected. The luciferase activity was determined 48 h after transfection using a dual-luciferase reporter assay system (Promega). Cell lysates (35 μL per well) were used for measurement of luciferase activity in a luminometer by first mixing the cell lysates (25 μL) with 20 μL luciferase assay reagent for measuring firefly luciferase activity and subsequently adding 20 μL Stop-Glo reagent for measuring Renilla luciferase activity. Data were normalized to Renilla luciferase activity.

**Cell growth and survival assay**. C4-2B MDVR or CWR22Rv1 cells were seeded on 12-well plates at a density of $0.3 \times 10^5$ cells per well in RPMI 1640 media containing 10% FBS and transfected with HSP70 siRNA or STUB1 plasmid and then treated with 20 μM enzalutamide or 5 μM abiraterone for 3 days. Total cell numbers were determined at 0, 3, and 6 days. CWR22Rv1 cells and C4-2B MDVR cells were seeded on 12-well plates at a density of $0.5 \times 10^5$ cells per well in RPMI 1640 media containing 10% FBS and treated with 5 μM APO or VER with or

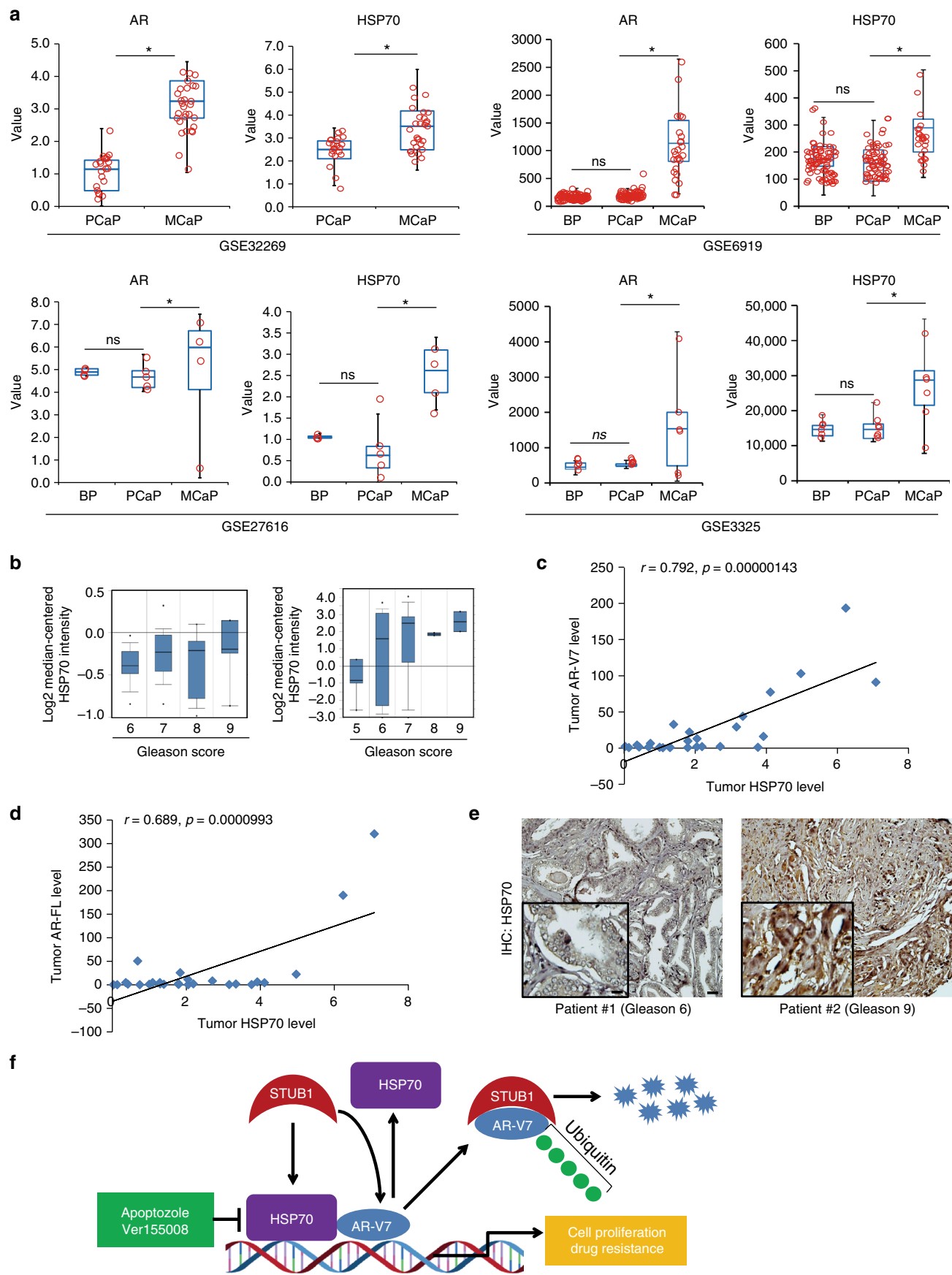

**Fig. 7** HSP70 level is correlated with AR/AR-V7 in prostate tumors. **a** In four independent GEO data bases (GSE32269, GSE6919, GSE27616, and GSE3325), AR and HSP70 gene expression levels were determined in benign prostate (BP), primary prostate cancer (PCaP), and metastatic prostate cancer (MCaP) tumor samples. *$p < 0.05$. Statistical analysis was performed using two-tailed Student's $t$-test. **b** In two independent Oncomine databases, HSP70 gene expression was determined in different Gleason score prostate tumors. **c, d** Total RNA from 26 high Gleason score tumors was isolated and mRNA expression of AR-FL, AR-V7, and HSP70 was determined by qRT-PCR. The AR-FL/HSP70 correlation and AR-V7/HSP70 correlation were determined by Spearman rank correlation. The correlation coefficient was determined. **e** The prostate tumor biopsies from two prostate cancer patients (Gleason 6 and Gleason 9, respectively) were fixed and HSP70 immunohistochemistry staining of the tumor sections was determined. Scale bar 50 μm (outside) and 20 μm (inside). **f** Proposed pathway of HSP70/STUB1/AR-V7 complex in next generation anti-androgen resistance and prostate cancer progression. HSP70 forms complex with AR-V7 and increases AR-V7 transcriptional activity, STUB1 binds with HSP70 and disassociates HSP70 and AR-V7 binding, the ubiquitin ligase U-box domain of STUB1 binds with AR-V7 and promotes its ubiquitination and degradation. HSP70 inhibition by APO or VER promotes STUB1 and AR-V7 binding and ubiquitination

without 20 μM enzalutamide or 5 μM abiraterone in media containing FBS. Total cell numbers were counted after 3 and 6 days.

**Clonogenic assay.** CWR22Rv1 cells or C4-2B MDVR cells were treated with 5 μM APO or VER with or without 20 μM enzalutamide or 5 μM abiraterone. Cells were plated at equal density (800 cells per dish) in 60 mm dishes for 3 weeks; the medium was changed every 7 days. The colonies were rinsed with PBS before staining with 0.5% crystal violet/4% formaldehyde for 30 min and the number of colonies was counted.

**Real-time quantitative RT-PCR.** Total RNA was extracted using TriZOL reagent (Invitrogen). cDNA was prepared after digestion with RNase-free RQ1 DNase (Promega) and then subjected to real-time reverse transcription-PCR (RT-PCR) using Sso Fast Eva Green Supermix (Bio-Rad) according to the manufacturer's instructions[58]. Each reaction was normalized by co-amplification of actin. Triplicates of samples were run on default settings of a Bio-Rad CFX-96 real-time cycler. The primer sequences are shown in Supplementary Table 6.

**Co-immunoprecipitation assay.** Equal amounts of cell lysates (1500 μg) were immunoprecipitated using 1 μg of Flag antibody, HSP70 antibody, AR-V7 antibody, AR (N20) antibody, or STUB1 antibody with 50 μL of protein A/G agarose with constant rotation overnight. The immunoprecipitants were washed with 10 mM HEPES (pH 7.9), 1 mM EDTA, 150 mM NaCl, and 1% Nonidet P-40 twice with 1 mL each. The precipitated proteins were eluted with 30 μL of SDS–PAGE sample buffer by boiling for 10 min. The eluted proteins were electrophoresed on 8% SDS–PAGE, transferred to nitrocellulose membranes, and probed with indicated antibodies.

**Dual immunofluorescence assay.** $1 \times 10^4$ 293 cells were plated in four-well Nunc™ Lab-Tek™ II Chamber Slides and transfected with AR-V7, HSP70, and Flag-STUB1 for 3 days and then treated with 10 μM APO or VER for another 24 h. Cells were fixed with 4% paraformaldehyde, permeabilized with 0.5% Triton X-100, and incubated with 1% BSA to block nonspecific binding. Cells were incubated with anti-AR (N20, Santa Cruz Biotechnology) and anti-Flag antibodies (Sigma) overnight. Intracellular AR-V7 was visualized with FITC-conjugated secondary antibodies, Flag-STUB1 was visualized with Texas red conjugated secondary antibodies and nuclei were visualized with DAPI by all-in-one fluorescence microscope (BZ-X700).

**Proteasome activity assay.** $2 \times 10^6$ C4-2B parental, C4-2B AbiR, C4-2B MDVR, and CWR22Rv1 cells were harvested and homogenized with 0.5% NP-40 in PBS. The proteasome activity was determined by Proteasome Activity Fluorometric Assay Kit (Biovision, Catalog #K245-100). Briefly, AMC standard curve was prepared as described in the instruction. 10 μL samples plus 90 μL assay buffer were added into opaque white 96 microwell plate. 1 μL of the proteasome inhibitor was added into one of the paired wells and 1 μL of the proteasome substrate was added to all the wells. The kinetics of fluorescence was developed at Ex/Em = 350/440 nm in a microplate reader (Molecular Devices, Llc.) at 37 °C for 0–120 min. ΔRFU = RFU (without proteasome inhibitor) – iRFU (with proteasome inhibitor). Proteasome activity = AMC amount of samples/(ΔRFU at linear time point 2 reading – ΔRFU at linear time point 1 reading) × sample volume × sample dilution factor = nmol per min per ml.

**RNA sequence data analysis.** C4-2B MDVR and CWR22Rv1 cells were treated with vehicle or the HSP70 inhibitors APO (10 μM) and VER (10 μM) for 48 h before RNA extraction. RNA-seq libraries from 1 μg total RNA were prepared using Illumina Tru-Seq RNA Sample, according to the manufacturer's instructions. mRNA-Seq paired-end library was prepared through Illumina NGS on HiSeq 4000: $2 \times 150$ cycles per bases (150 bp, PE). Around 30M reads per sample were generated. Data analysis was performed with a Top Hat–Cufflinks pipeline and sequence read mapping/alignment using HISAT. StringTie Data was mapped to and

quantified for 60,658 unique genes/transcripts Gene and transcript expression is quantified as Fragments Per Kilobase of transcript per Million mapped reads (FPKM). Principal component analysis (PCA) was conducted on the FPKM gene-level data for all genes/transcripts passing filter (filtered on expression > 0.1) in the raw data. The relatedness of the differentially expressed genes from APO and VER treatment was depicted with a Venn diagram. The common regulated genes by APO and VER treatment were clustered with Hierarchical Clustering algorithm by StrandNGS software.

**Gene set enrichment analysis (GSEA).** GSEA was performed using the Java desktop software (http://software.broadinstitute.org/gsea/index.jsp)[59]. Genes were ranked according to the shrunken limma $\log_2$ fold changes, and the GSEA tool was used in 'pre-ranked' mode with all default parameters. KEGG-Ubiquitin-mediated proteolysis pathway was used in the GSEA analysis.

**Ingenuity pathway analysis.** Pathway analysis of transcripts with changed expression in C4-2B parental and C4-2B MDVR cells was performed using IPA (www.ingenuity.com) and canonical pathways were determined. The $p$-value associated with a pathway is a measure of the likelihood that the association between a set of focus genes in the experiment and a given process or pathway is the result of random chance; in general, a $p$-value (calculated using the right-tailed Fisher exact test) < 0.05 indicates a statistically significant value.

**Gene expression omnibus (GEO) and Oncomine analysis.** Four separate data sets from NCBI GEO were screened independently for expression levels of AR and HSP70. GSE32269 compared localized primary prostate cancer and metastatic prostate cancer. GSE6919, GSE27616, and GSE3325 compared benign prostate specimens, primary prostate tumor, and metastatic prostate cancer. Data generated from two prostate carcinoma data sets (Lapointe prostate and Singh prostate) from Oncomine were analyzed and HSP70 expression in different Gleason score tumors was determined.

**Docking and binding analysis with Autodock/Vina and PyMOL.** The X-ray crystallographic structure of HSP70 ATPase domain was retrieved from RCSB Database (PDB code: 1S3X). The Ligands (APO and VER) preparation and grid box creation were completed using Graphical User Interface program AutoDock Tools (ADT). AutoDock saved the prepared file in PDBQT format. AutoGrid was used for the preparation of the grid map using a grid box. The grid size was set to $50 \times 50 \times 50$ $xyz$ points with grid spacing of 0.375 Å and vina search space center was designated at dimensions ($x$, $y$, and $z$): 17.3821, 28.7233, and 16.0554. Auto-Dock/Vina was employed for docking using protein and ligands information along with grid box properties in the configuration file. During the docking procedure, both the protein and ligands are considered as rigid. The results <1.0 Å in positional root-mean-square deviation (RMSD) was clustered together and represented by the result with the most favorable free energy of binding. The pose with lowest energy of binding or binding affinity was extracted and aligned with receptor structure. The obtained docked poses were analyzed with ADT using PyMOL.

**Human prostate cancer patient sample analysis.** The tumor samples were harvested through the University of California, Davis (UC Davis) central bior-epository under an approved Institutional Review Board (IRB) protocol in Urology Department (#GU001). The informed consent was obtained from all participating patients. Biopsy samples were taken from patients and immediately frozen into liquid nitrogen before the RNA extraction. Twenty-six tumors were classified as high Gleason score (≥8) groups, including 13 HSPC (hormone sensitive prostate cancer) and 13 CRPC (castration-resistant prostate cancer) samples (Supplementary Table 1). In the 13 CRPC samples, nine were from prostate biopsies, two from bone biopsies, and two from lymph node biopsies. Total RNA from these patient samples was extracted by RNeasy mini plus kit (QIAGEN) and cDNA was prepared after digestion with RNase-free RQ1 DNase (Promega). qRT-PCR was run using Sso Fast Eva Green Supermix (Bio-Rad) according to the manufacturer's

instructions and as described. AR-FL, AR-V7, HSP70, and HSP90 expression levels were determined. The primer sequences are shown in Supplementary Table 6.

**Conditionally reprogramed cells (CRCs) culture.** Primary cells from malignant human prostate tissues were isolated according to the protocol[30]. Briefly, human prostate tissues were minced and digested with collagenase/hyaluronidase/dispase at 37 °C for 1–3 h. The dissociated cell suspension was filtered through a 100 μM cell strainer and collected. Cells were plated with mixtures of complete F medium/conditioned medium of irradiated J2 culture, supplemented with 10 μM Y-27632. Subculturing was performed with trypsin treatments when needed.

**Animal studies and treatment regimens.** All animals used in this study received humane care in compliance with applicable regulations, policies, and guidelines relating to animals. All experimental procedures using animals were approved by the Institutional Animal Care and Use Committee of UC Davis. CWR22Rv1 cells (3 million) were mixed with matrigel (1:1) and injected subcutaneously into the flanks of 4–5-week-old male C.B17/lcrHsd-Prkdc-SCID mice (ENVIGO). Tumor-bearing mice (tumor volume around 50–100 mm$^3$) were randomized into six groups (seven mice per group) and treated as follows: (1) vehicle control (15% Cremophor EL, 82.5% PBS, and 2.5% dimethyl sulfoxide (DMSO), intraperitoneal (i.p.)), (2) enzalutamide (25 mg/kg, per os (p.o.)), (3) APO (5 mg/kg, i.p.), (4) VER (15 mg/kg i.p.), (5) enzalutamide (25 mg/kg, p.o.) plus APO (5 mg/kg, i.p.), and (6) enzalutamide (25 mg/kg, p.o.) plus VER (15 mg/kg, i.p.). Tumors were measured using calipers twice a week and tumor volumes were calculated using length × width × width × 0.52. Tumor tissues, liver, and kidney were harvested and weighed after 3 weeks of treatment. Tumor tissues, liver, and kidney were paraffin embedded and H/E stained.

To assess the effect of combination of HSP70 inhibitors with enzalutamide on the growth of PDX tumors, the LuCaP35 CR model was obtained from the University of Washington and established in the UC Davis Cancer Center. Briefly, 3–4 weeks C.B17/lcrHsd-Prkdc-SCID mice (ENVIGO) were surgically castrated. Two weeks later, ~20 to 30-mm$^3$ pieces of LuCaP 35CR tumor were implanted into the pre-castrated SCID mice. When tumors reached 50–100 mm$^3$, mice were randomized into four groups (six mice per group) and treated as follows: (1) vehicle control (15% cremophor EL, 82.5% PBS, and 2.5% DMSO, i.p.), (2) enzalutamide (25 mg/kg, p.o.), (3) APO (5 mg/kg, i.p.), and (4) enzalutamide (25 mg/kg, p.o.) plus APO (5 mg/kg, i.p.). Tumors were measured using calipers twice a week and tumor volumes were calculated using length × width × width × 0.52. Tumor tissues were harvested and weighed after 5 weeks of treatment. Serum was collected for PSA determination.

**Measurement of mouse serum PSA.** Mouse blood from the LuCaP 35CR tumor model was collected and the serum was harvested. PSA levels were measured using a PSA ELISA Kit (United Biotech, Inc., Mountain View, CA) according to the manufacturer's instructions.

**Immunohistochemistry.** Tumors were fixed by formalin and paraffin-embedded tissue blocks were dewaxed, rehydrated, and blocked for endogenous peroxidase activity. Antigen retrieving was performed in sodium citrate buffer (0.01 mol per Litter, pH 6.0) in a microwave oven at 1000 W for 3 min and then at 100 W for 20 min. Nonspecific antibody binding was blocked by incubating with 10% FBS in PBS for 30 min at room temperature. Slides were then incubated with anti-HSP70 (F-3, at 1:300; Santa Cruz), anti-Ki67 (at 1:500; Neomarker), or anti-AR-V7 (at 1:200; Precision) at 4 °C overnight. Slides were then washed and incubated with biotin-conjugated secondary antibodies for 30 min, followed by incubation with avidin DH-biotinylated horseradish peroxidase complex for 30 min (Vectastain ABC Elite Kit, Vector Laboratories). The sections were developed with the diaminobenzidine substrate kit (Vector Laboratories) and counterstained with hematoxylin. Nuclear staining of cells was scored and counted in five different vision fields. Images were taken with an Olympus BX51 microscope equipped with DP72 camera.

**Statistical analysis.** Statistical analyses were performed with SPSS16.0. Raw data was summarized by means, standard deviations (SD), and graphical summaries and transformed if necessary to achieve normality. Data from the in vitro experiments are presented as means ± SD from three independent experiments. Differences between individual groups were analyzed by two-tailed Student's t tests for single comparisons or one-way analysis of variance (ANOVA) followed by the Scheffé procedure for multiple group comparisons. In the tumor growth experiments, size of the tumor at sacrifice serves as the primary response measure. The tumor growth and PSA across groups was analyzed by ANOVA. Concordance between AR-FL, AR-V7, HSP70, and HSP90 level in clinical patient samples was determined by Spearman rank correlation. $p < 0.05$ was considered statistically significant.

## Data availability

The RNA sequence data and Microarray data in the present study have been deposited in Gene Expression Omnibus (GEO) with the accession number GSE120006. The hyperlink of the dataset is: https://www.ncbi.nlm.nih.gov/geo/query/acc.cgi?acc=GSE120006. All data are available from the authors upon reasonable request.

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

## Acknowledgements

We sincerely thank the Genomics Shared Resource (GSR) and Biorepository Shared Resource (BSR) at the UC Davis Medical Center for their assistance in our study. We are grateful for the technical support by Dr. Yoshihiro Izumiya at the Department of Dermatology, UC Davis Medical Center. This work was supported in part by grants NIH/NCI CA168601, CA179970, DOD PC150229, and the U.S. Department of Veterans Affairs, Office of Research & Development BL&D grant number I01BX0002653 (A.C.G.), a Research Career Scientist Award (A.C.G.). A.C.G. is also a Research Career Scientist at VA Northern California Health Care System, Mather, California.

## Author contributions

C.F.L. and A.C.G. conceived the project and designed the experiments. C.F.L., W.L., J.C.Y., and A.C.G. developed the methodology. C.F.L., W.L., J.C.Y., L.R.L., and R.N.Z. performed the experiments and acquired the data. C.P.E., O.D.N., H.W.C., and M.D. coordinated the clinical specimen acquisition and provided technical and material support. C.F.L. and C.G.T. performed the bioinformatics analysis. C.F.L., J.C.Y., C.P.E., and A.C.G. interpreted and analyzed the data. C.F.L., J.C.Y., and A.C.G. wrote the manuscript. C.M.A. and A.P.L. edited the manuscript. C.F.L. and A.C.G. supervised the study.

## Additional information

**Competing interests:** The authors declare no competing interests.

