## [Peer Review File · Nature Communications]

Reviewers' Comments:

Reviewer #1:

Remarks to the Author:

In this manuscript (#NCOMMS-18-13549), Dr. Gao and colleagues discovered that prostate cancer cells resistant to anti-androgen therapies display suppressed ubiquitin-proteasome system (UPS), which is associated with mitigated proteasomal degradation of AR and AR-V7 variants, key players in conferring resistance to anti-androgen therapies. In particular, the authors discovered that while the HSP70/HSC70-interacting E3 ligase STUB1/CHIP is down-regulated in these anti-androgen resistant prostate cancer cells, HSP70/HSC70 is up-regulated, correlating with increased AR/AR-V7 protein levels. Mechanistically, the authors demonstrated that STUB1/CHIP mediates the ubiquitination of AR/AR-V7 proteins and subsequent proteasomal degradation, which is antagonized by HSP70. Conversely, either genetic or pharmacological inhibition of HSP70 leads to enhanced AR/AR-V7 ubiquitination and degradation. Importantly, pharmacological HSP70 inhibitors markedly sensitize anti-androgen resistant prostate cancer cells to enzalutamide and abiraterone treatment both in vitro and in vivo. Thus, the authors proposed inhibition of HSP70 as a potential therapeutic strategy to reduce AR-V7 expression and overcome resistance to AR-targeted therapies, a notion that is certainly interesting to the prostate cancer research community.

This manuscript presents a great deal of data, which largely support the major conclusion overall—that is, HSP70 overexpression contributes to AR/AR-V7 stability by preventing their ubiquitination and proteasomal degradation. Nonetheless, a number of issues, which dampen the reviewer's enthusiasm, need to be carefully addressed before being considered for publication. Thus, a major revision is recommended by the reviewer.

Major concerns:

1. The major findings presented in this manuscript are not highly original. Previously, it has been known that STUB1/CHIP can mediate AR/AR-V7 polyubiquitination and degradation (Biochim Biophys Acta. 2006 Jun;1764(6):1073-9.; EMBO J. 2011 Feb 2;30(3):468-79; Oncogene. 2014 Jan 2;33(1):26-33). Moreover, HSP70 inhibitors have been shown to cause rapid AR/AR-V7 protein degradation and impede the in vivo growth of castration-resistant prostate cancer cells in xenograft models (Cancer Res. 2018 May 15. pii: canres.3728.2017. doi: 10.1158/0008-5472).
2. There are several data inconsistencies. While the protein level of AR-V7 is markedly elevated in CWR22Rv1 cells compared to C4-2B cells, the amount of precipitated AR-V7 is less in this cell line (Fig. 1h). In addition, the immunoblots of precipitated AR-V7 and AR-FL are overexposed (Fig. 1h), which makes it difficult to judge equal precipitation. Similarly, in Fig. 2g overexpression of FLAG-STUB1 is expected to cause increased overall protein ubiquitination; in contrast, the inputs show reduced protein ubiquitination. More importantly, the ubiquitination of AR-V7 is only slightly increased in STUB1-overexpressing cells (Fig. 2g), which is not convincing. In Fig. 3b, overexpression of HSP70, given the huge number of HSP70 substrates, is expected to reduce overall protein ubiquitination; however, the inputs do not show obvious changes in protein ubiquitination. In contrast, the ubiquitination of AR-V7 is markedly reduced (Fig. 3b), implying that HSP70 selectively suppresses AR-V7 ubiquitination. Again, in Fig. 3j HSP70 inhibitors are expected to induce overall protein ubiquitination; however, the inputs show reduced ubiquitination. In contrast, AR-V7 ubiquitination is increased. This selectivity would be very surprising and contrast with the well-known impact of HSP70 as a pivotal chaperone on the whole proteome. This important point should be further investigated and elucidated.
3. Throughout the manuscript, the nomenclatures of key genes/proteins are not clearly defined and confusing. In mammals there are 13 HSP70 family members, among which HSPA1A and HSPA1B encode the stress-inducible HSP72 and HSPA8 encodes the constitutively expressed HSC70/HSP73. The authors do not define which HSP70 family member was overexpressed or

detected in their experiments (Fig. 2c, 2h, 3a, 3b, 3k, and 6). For example, in Fig. 3a which HSP70 family member was detected, HSPA1A, HSPA1B or HSPA8? More importantly, in Fig. 3e which HSP70 isoforms were knocked down, the inducible, constitutive form or both?

4. In Fig. 3e, the authors should use a dual luciferase reporter system, since luciferase folding requires HSP70. Thus, the reduced PSA reporter activities may be simply due to impaired luciferase folding and/or protein translation following HSP70 knockdown.

5. As HSP70 is an essential chaperone required for the viability of every cell type including primary cells, it is not surprising that HSP70 inhibition impair the growth of prostate cancer cells (Fig. 3c, 3d, 3f and 3h). Rather, it is important to demonstrate the differential responses between tumor cells and primary cells. The authors should include primary cells for comparison in these experiments.

6. While the authors focused on stabilization of AR-V7 by HSP70, it was previously reported that the HSP70 inhibitor VER 155008 decreases AR/AR-V7 mRNA levels (Cancer Sci. 2017 Sep;108(9):1820-1827). Thus, the reduction in AR-V7 protein levels following HSP70 inhibition may also be partially due to impaired AR-V7 splicing. The authors should also examine the mRNA levels of AR-V7 following HSP70 inhibition.

7. In addition to STUB1/CHIP, MDM2 is another known E3 ligase for AR/AR-V7 (EMBO J. 2002 Aug 1;21(15):4037-48). Although the authors showed enhanced STUB1-AR-V7 interactions upon HSP70 inhibition (Fig. 3k), it remains unknown whether MDM2-AR-V7 interactions are also enhanced and, more importantly, whether STUB1/CHIP plays a predominant role or not under this condition. To address this question, the authors should knock down STUB1/CHIP in the context of HSP70 inhibition to see if the AR-V7 level is largely or markedly restored.

8. Given the very broad effects of HSP70 on the proteome, AR-V7 is just one of the numerous targets affected by HSP70 inhibition. Thus, it is important to demonstrate how much the AR-V7 degradation contributes to the tumor suppression caused by HSP70 inhibition. The authors should express a degradation-resistant mutant or simply overexpress wild-type AR-V7 in prostate cancer cells treated with HSP70 inhibitors to see if the tumor growth can be rescued. Otherwise, HSP70 inhibition, which would cause decrease in overall cell fitness, can just sensitize any kind of therapies.

Minor concerns:

1. Throughout the manuscript, the statistical methods are described very briefly. Clearly, not all data are presented as mean \pm SD, such as Fig. 6. Also, not all analyses use ANOVA, such as Fig. 6a (only two groups).

2. Throughout the manuscript, HSP70 and HSP90 isoforms should be specified and appropriate nomenclature should be used.

3. In Fig. 1h, Fig. 2g and Fig. 3b (right panel), it should be labeled as IB: Ub, not Ub-Ar-V7, as in the inputs the ubiquitin Ab detects global protein ubiquitination.

4. In Fig. 3h, the SICON cell line should be DMSO?

5. In Fig. 5b, there are 7 mice each group. However, in the methods it says 6 mice per group.

Reviewer #2:

Remarks to the Author:

In this manuscript, the authors show changes in protein homeostasis correlated with resistance to AR-targeting drugs. They started from the observation that the ubiquitination system is suppressed in different enzalutamide and abiraterone resistant cell models. They conclusively show that AR and AR-Vs expression levels are regulated by STUB1/hsp70 complex. STUB1 and hsp70 physically interact with each other and with AR in expressing cell lines. It is demonstrated that inhibiting hsp70 reinstals enzalutamide sensitivity in these cellular model, correlated with changed AR-V levels. This is also shown in CRPC xenograft models. Clinical relevance is validated in gene expression of high Gleason scoring prostate tumors where hsp70 is correlated with AR and AR-V expression.

The experiments are well described in sufficient detail, executed in different cellular models, the result section is clear, discussion is inclusive of current literature and knowledge.

In conclusion, a well designed and executed work.

We appreciate the comments by the reviewers and have incorporated their suggestions in the revised manuscript. The following responses address the reviewers' comments in a point-by-point manner.

Reviewer #1

In this manuscript (#NCOMMS-18-13549), Dr. Gao and colleagues discovered that prostate cancer cells resistant to anti-androgen therapies display suppressed ubiquitin-proteasome system (UPS), which is associated with mitigated proteasomal degradation of AR and AR-V7 variants, key players in conferring resistance to anti-androgen therapies. In particular, the authors discovered that while the HSP70/HSC70-interacting E3 ligase STUB1/CHIP is down-regulated in these anti-androgen resistant prostate cancer cells, HSP70/HSC70 is up-regulated, correlating with increased AR/AR-V7 protein levels. Mechanistically, the authors demonstrated that STUB1/CHIP mediates the ubiquitination of AR/AR-V7 proteins and subsequent proteasomal degradation, which is antagonized by HSP70. Conversely, either genetic or pharmacological inhibition of HSP70 leads to enhanced AR/AR-V7 ubiquitination and degradation. Importantly, pharmacological HSP70 inhibitors markedly sensitize anti-androgen resistant prostate cancer cells to enzalutamide and abiraterone treatment both in vitro and in vivo. Thus, the authors proposed inhibition of HSP70 as a potential therapeutic strategy to reduce AR-V7 expression and overcome resistance to AR-targeted therapies, a notion that is certainly interesting to the prostate cancer research community.

This manuscript presents a great deal of data, which largely support the major conclusion overall-that is, HSP70 overexpression contributes to AR/AR-V7 stability by preventing their ubiquitination and proteasomal degradation. Nonetheless, a number of issues, which dampen the

reviewer's enthusiasm, need to be carefully addressed before being considered for publication. Thus, a major revision is recommended by the reviewer.

Major concerns:

1. The major findings presented in this manuscript are not highly original. Previously, it has been known that STUB1/CHIP can mediate AR/AR-V7 polyubiquitination and degradation (Biochim Biophys Acta. 2006 Jun;1764(6):1073-9.; EMBO J. 2011 Feb 2;30(3):468-79; Oncogene. 2014 Jan 2;33(1):26-33). Moreover, HSP70 inhibitors have been shown to cause rapid AR/AR-V7 protein degradation and impede the in vivo growth of castration-resistant prostate cancer cells in xenograft models (Cancer Res. 2018 May 15. pii: canres.3728.2017. doi: 10.1158/0008-5472).

Answer: Our manuscript systematically investigated the role of the HSP70/STUB1 complex in next generation anti-androgen resistant prostate cancer (such as enzalutamide and abiraterone resistant). The significance of our manuscript is to provide a novel mechanism of next generation anti-androgen resistance via ubiquitin-proteasome-system (UPS) alteration and identify a regulatory mechanism of androgen receptor variants (especially AR-V7) proteostasis mediated by the STUB1/HSP70 complex. We established the novel drug resistant models and discovered that the ubiquitin mediated proteolysis pathway and proteasome activity are suppressed in enzalutamide and abiraterone resistant prostate cancer cells. HSP70 inhibition activated Unfolded Protein Response (UPR) and significantly disrupts AR/AR-V7 gene programs. Additionally, we showed that the levels of HSP70 are correlated with AR-V7 in tumors from patients with high Gleason scores. Our findings suggested that the chaperone-ubiquitin-proteasome alteration may represent a general mechanism for the regulation of AR variants

protein stability and we provided the rationale to target proteostasis through inhibition of HSP70 as a potential therapeutic strategy to overcome the resistance to the AR-targeted therapies in castration resistant prostate cancer (CRPC) patients.

In the revised manuscript, we provided additional data to support that STUB1/HSP70 complex may represent a novel mechanism conferring next generation anti-androgen resistance through the regulation of AR and AR variants (such as AR-V1, AR-V3, AR-V7, AR-V9 and AR567^{es}) proteostasis. We provided co-immunoprecipitation data to show that STUB1 and HSP70 bind with AR-V1, AR-V3, AR-V7, AR-V9 and AR567^{es} (**Fig.2c**), suggesting STUB1/HSP70 complex might represent a general mechanism to regulate proteostasis of all the AR variants. We also provided the STUB1 knockdown data and dual immunofluorescence staining data to support the important role of STUB1 involved in AR-V7 protein degradation by HSP70 inhibition (**Fig.4**). Furthermore, we established the Conditionally Reprogrammed cells (CRCs) from a Gleason 10 prostate cancer patient and found that AR and HSP70 were overexpressed in the CRCs by immunofluorescence staining. We also tested their response to enzalutamide, abiraterone and HSP70 inhibitors (APO and VER). The patient derived CRCs showed resistance to both enzalutamide and abiraterone; however, APO and VER can suppress cell proliferation in a dose dependent manner. Moreover, combined APO or VER with enzalutamide further suppressed the CRCs growth (**Supplementary Fig.6**). We also added more discussion about our novel findings involved in the next generation antiandrogen resistance models in the revised manuscript (**Page 16**).

2. There are several data inconsistencies. While the protein level of AR-V7 is markedly elevated in CWR22Rv1 cells compared to C4-2B cells, the amount of precipitated AR-V7 is less in this cell line (Fig. 1h). In addition, the immunoblots of precipitated AR-V7 and AR-FL are

overexposed (Fig. 1h), which makes it difficult to judge equal precipitation. Similarly, in Fig. 2g overexpression of FLAG-STUB1 is expected to cause increased overall protein ubiquitination; in contrast, the inputs show reduced protein ubiquitination. More importantly, the ubiquitination of AR-V7 is only slightly increased in STUB1-overexpressing cells (Fig. 2g), which is not convincing. In Fig. 3b, overexpression of HSP70, given the huge number of HSP70 substrates, is expected to reduce overall protein ubiquitination; however, the inputs do not show obvious changes in protein ubiquitination. In contrast, the ubiquitination of AR-V7 is markedly reduced (Fig. 3b), implying that HSP70 selectively suppresses AR-V7 ubiquitination. Again, in Fig. 3j HSP70 inhibitors are expected to induce overall protein ubiquitination; however, the inputs show reduced ubiquitination. In contrast, AR-V7 ubiquitination is increased. This selectivity would be very surprising and contrast with the well-known impact of HSP70 as a pivotal chaperone on the whole proteome. This important point should be further investigated and elucidated.

Answer: We have repeated the **Fig.2g** experiments. As shown in Fig.2g in the revised manuscript, STUB1 significantly increased AR-V7 ubiquitination and adding MG132 further increased the AR-V7 ubiquitination by STUB1.

We have replaced **Fig.1h** with improved pictures to show AR-V7/AR-FL is equally precipitated in the revised manuscript.

We agree with the reviewer that HSP70 functions as a pivotal chaperone on the whole proteome and it must be regulating many oncogenic proteins in cancer cells. However, the Fig.2g, Fig.3b and Fig.3j experiments were performed in HEK293 cells co-transfected with the HA-Ub plasmid. Thus, the ubiquitin detected in the input was the exogenous overexpressed HA-Ub and it may be

hard to reflect the changes the reviewer indicated. This is also confirmed in other publications (*Paul I et al. Oncogene. 2013 Mar 7;32(10):1284-95. Yu Z et al., Elife. 2016 Apr 11;5. pii: e14087.*). Additionally, we examined the endogenous ubiquitin expression change with HSP70 inhibition in the revised manuscript. C4-2B MDVR cells were treated with APO and VER for 24 hours and then treated with 5 μ M MG132 for another 6 hours. Total cell lysates were collected and the total ubiquitin was determined by western blot. As shown in **Supplementary Fig.4h**, without MG132, APO and VER slightly increased total ubiquitination; however, after the proteasome activity was inhibited by MG132, ubiquitination was significantly increased by APO and VER treatment. These results suggested APO and VER induced a global change of the whole cell ubiquitination and HSP70 functions as a pivotal chaperone on the whole proteome.

3. Throughout the manuscript, the nomenclatures of key genes/proteins are not clearly defined and confusing. In mammals there are 13 HSP70 family members, among which HSPA1A and HSPA1B encode the stress-inducible HSP72 and HSPA8 encodes the constitutively expressed HSC70/HSP73. The authors do not define which HSP70 family member was overexpressed or detected in their experiments (Fig. 2c, 2h, 3a, 3b, 3k, and 6). For example, in Fig. 3a which HSP70 family member was detected, HSPA1A, HSPA1B or HSPA8? More importantly, in Fig. 3e which HSP70 isoforms were knocked down, the inducible, constitutive form or both?

Answer: All of the overexpressed HSP70 (Fig. 2c, 2h, 3a, 3b, 3k) are HSPA1B in the manuscript. The plasmid is from OriGene (Catalog# SC116767). The two independent HSP70 siRNA were from Invitrogen (Catalog# 262305 and 262306), the specific genes targeted for knockdown are HSPA1A/HSPA1B. We have modified the text in the revised manuscript to specify this (**Page 4 and 24**).

4. In Fig. 3e, the authors should use a dual luciferase reporter system, since luciferase folding requires HSP70. Thus, the reduced PSA reporter activities may be simply due to impaired luciferase folding and/or protein translation following HSP70 knockdown.

Answer: A dual luciferase reporter system was used in the manuscript as previously described in our earlier publications (*Clin Cancer Res.* 2014;20(12):3198-3210.). We have described it in greater detail in the revised methods (**Page 25**).

5. As HSP70 is an essential chaperone required for the viability of every cell type including primary cells, it is not surprising that HSP70 inhibition impair the growth of prostate cancer cells (Fig. 3c, 3d, 3f and 3h). Rather, it is important to demonstrate the differential responses between tumor cells and primary cells. The authors should include primary cells for comparison in these experiments.

Answer: We appreciate the reviewer's suggestion. In the revised manuscript, we have included data from primary normal fibroblast cells IMR90, and the immortalized normal prostate epithelial cell line PZ-HPV7, both of human origin. As shown in revised **Fig.3e** and **Supplementary Fig.3b-d**, IMR90 and PZHPV7 expressed low levels of HSP70 but did not express AR or AR-V7. APO and VER treatment slightly suppressed PZ-HPV7 cell growth and had no effects on IMR90 cells. Knockdown of HSP70 in PZ-HPV7 or IMR90 did not affect cell growth or enzalutamide sensitivity. We also tested the HSP70 inhibitors and anti-androgens combination in these two cell lines. As shown in **Supplementary Fig.5b**, APO, VER, enzalutamide and abiraterone treatment did not significantly suppress PZHPV7 and IMR90 cell

growth, and combination of HSP70 inhibitors with enzalutamide or abiraterone had no further effects on the cell growth. To further expand our study into clinical settings, we used the above-mentioned CRCs and treated them with APO and VER as well as enzalutamide and abiraterone. As shown in **Supplementary Fig.6**, the CRCs derived from the patient expressed significantly higher HSP70 and AR in the nucleus. Human mitochondrial staining confirmed that the cells were of human origin. We then treated the CRCs with different doses of APO and VER. These cells were resistant to enzalutamide and abiraterone treatment; however, APO and VER treatment suppressed cell proliferation in a dose dependent manner. Additionally, combining APO or VER with enzalutamide further suppressed cell proliferation.

6. While the authors focused on stabilization of AR-V7 by HSP70, it was previously reported that the HSP70 inhibitor VER 155008 decreases AR/AR-V7 mRNA levels (Cancer Sci. 2017 Sep;108(9):1820-1827). Thus, the reduction in AR-V7 protein levels following HSP70 inhibition may also be partially due to impaired AR-V7 splicing. The authors should also examine the mRNA levels of AR-V7 following HSP70 inhibition.

Answer: In the revised manuscript, we analyzed AR variants mRNA expression by qRT-PCR. C4-2B MDVR cells were treated with APO and VER. We determined the APO and VER effects on mRNA level of AR-FL/AR variants and their targets genes in C4-2B MDVR cells (**Supplementary Fig.4a-g**). Intriguingly, APO slightly decreased AR-V7 mRNA level, but significantly decreased AR-FL, AR-V1, AR-V3 and AR-V9 mRNA level. VER significantly decreased AR-V3 and AR-V9 mRNA expression but not AR-FL and AR-V7 mRNA level. These results suggested APO and VER treatment may regulate AR variants expression by other mechanisms. However, proteasome inhibitor MG132 largely recused the effects of APO and

VER on AR/AR-V7 suppression (**Supplementary Fig.4h**), indicating APO and VER suppress AR-V7 expression largely through protein degradation.

7. In addition to STUB1/CHIP, MDM2 is another known E3 ligase for AR/AR-V7 (EMBO J. 2002 Aug 1;21(15):4037-48). Although the authors showed enhanced STUB1-AR-V7 interactions upon HSP70 inhibition (Fig. 3k), it remains unknown whether MDM2-AR-V7 interactions are also enhanced and, more importantly, whether STUB1/CHIP plays a predominant role or not under this condition. To address this question, the authors should knock down STUB1/CHIP in the context of HSP70 inhibition to see if the AR-V7 level is largely or markedly restored.

Answer: Our study established a novel model where the HSP70/STUB1 complex controls next generation anti-androgen resistance through AR/AR-V7 proteostasis. To further confirm that HSP70 inhibition degrades AR-V7 through STUB1, we first confirmed our findings that APO and VER treatment significantly promotes STUB1 and AR/AR-V7 binding in C4-2B MDVR cells as well as in the 293 cell system by dual immunofluorescence staining (**Fig.4a-c**). Additionally, APO and VER treatment significantly suppressed AR/AR-V7 expression. However, knockdown of STUB1 abolished the APO and VER effects on AR/AR-V7, suggesting suppression of AR/AR-V7 by APO and VER is predominately through STUB1 (**Fig.4d**).

We also agree with the reviewer's comment that MDM2 is another important E3 ubiquitin ligase that involves in AR/AR-V7 protein degradation. Although in depth analysis of MDM2 is beyond the scope of our manuscript, we measured MDM2 expression in enzalutamide/abiraterone

resistant cells and found that MDM2 expression is not significantly changed in all resistant CRPC cell lines (**Fig.3a**).

8. Given the very broad effects of HSP70 on the proteome, AR-V7 is just one of the numerous targets affected by HSP70 inhibition. Thus, it is important to demonstrate how much the AR-V7 degradation contributes to the tumor suppression caused by HSP70 inhibition. The authors should express a degradation-resistant mutant or simply overexpress wild-type AR-V7 in prostate cancer cells treated with HSP70 inhibitors to see if the tumor growth can be rescued. Otherwise, HSP70 inhibition, which would cause decrease in overall cell fitness, can just sensitize any kind of therapies.

Answer: We agree with the reviewer that HSP70 may be a master regulator that could control proteostasis of different oncogenic proteins. We believe that AR-V7 is one of them. In the revised manuscript, we determined that AR-V7 and HSP70 co-overexpression in C4-2B cells can greatly rescued the growth inhibition induced by APO and VER. As shown in **Fig.4e-f**, overexpression of both HSP70 and AR-V7 into C4-2B cells in CS-FBS conditions significantly increased cell growth. APO and VER still suppressed cell growth in HSP70/AR-V7 overexpressing cells; however, the proliferation rate in the HSP70/AR-V7 overexpressed cells was higher than that in the vector alone and single HSP70 or AR-V7 transfected cells. These results suggested that HSP70 stabilized and protected AR-V7 from degradation. Inhibition of HSP70 suppressed cell growth through the AR-V7 regulation.

Additionally, we provided data to support that inhibition of HSP70 or overexpression of STUB1 decreased glucocorticoid receptor (GR) transcriptional activity which confirm that HSP70

functions as a pivotal chaperone on the activation of different nuclear receptors (**Supplementary Fig.3i-j**).

Minor concerns:

1. Throughout the manuscript, the statistical methods are described very briefly. Clearly, not all data are presented as mean \pm SD, such as Fig. 6. Also, not all analyses use ANOVA, such as Fig. 6a (only two groups).

Answer: We appreciate the reviewer's correction and we have modified the statistical methods section and added more details in the revised manuscript accordingly (**Page 31**).

2. Throughout the manuscript, HSP70 and HSP90 isoforms should be specified and appropriate nomenclature should be used.

Answer: We appreciate the reviewer's suggestion and we have specified the HSP70 and HSP90 isoforms in the revised manuscript (**Page 4,8 and 24**).

3. In Fig. 1h, Fig. 2g and Fig. 3b (right panel), it should be labeled as IB: Ub, not Ub-Ar-V7, as in the inputs the ubiquitin Ab detects global protein ubiquitination.

Answer: We appreciate the reviewer's suggestion and have modified the figures accordingly.

4. In Fig. 3h, the SICON cell line should be DMSO?

Answer: We appreciate the reviewer's correction.

5. In Fig. 5b, there are 7 mice each group. However, in the methods it says 6 mice per group.

Answer: We appreciate the reviewer's correction. We have 7 mice in CWR22Rv1 xenograft tumor groups and 6 mice in LuCaP35 CR PDX tumor groups. We have modified the methods accordingly.

Reviewer #2

In this manuscript, the authors show changes in protein homeostasis correlated with resistance to AR-targeting drugs. They started from the observation that the ubiquitination system is suppressed in different enzalutamide and abiraterone resistant cell models. They conclusively show that AR and AR-Vs expression levels are regulated by STUB1/hsp70 complex. STUB1 and hsp70 physically interact with each other and with AR in expressing cell lines. It is demonstrated that inhibiting hsp70 reinstals enzalutamide sensitivity in these cellular model, correlated with changed AR-V levels. This is also shown in CRPC xenograft models. Clinical relevance is validated in gene expression of high Gleason scoring prostate tumors where hsp70 is correlated with AR and AR-V expression.

The experiments are well described in sufficient detail, executed in different cellular models, the result section is clear, discussion is inclusive of current literature and knowledge. In conclusion, a well designed and executed work.

Answer: We appreciate the reviewer's comment.

Reviewers' Comments:

Reviewer #1:

Remarks to the Author:

The authors have addressed most of the reviewer's concerns satisfactorily, except one minor point, likely a technical issue. In revised Fig. 2g, without MG132 treatment, while the input shows significantly reduced total AR-V7, due to STUB1-mediated degradation, the IP results indicate equal amounts of IPed AR-V7, which may be due to insufficient Abs used for IP. Thus, this issue needs to be fixed before formal publication.

We are glad to hear that our manuscript is principally accepted for publication in Nature Communications once it is revised in the compiled format. The following response addresses the reviewer's comment.

Reviewer #1 (Remarks to the Author):

The authors have addressed most of the reviewer's concerns satisfactorily, except one minor point, likely a technical issue. In revised Fig. 2g, without MG132 treatment, while the input shows significantly reduced total AR-V7, due to STUB1-mediated degradation, the IP results indicate equal amounts of IPed AR-V7, which may be due to insufficient Abs used for IP. Thus, this issue needs to be fixed before formal publication.

Answer: In Fig.2g, the input shows a significant reduction of total amounts of AR-V7 by STUB1 transfection without MG132 treatment. For the IP results, while the amounts of AR-V7 were equally precipitated, the levels of ubiquitination in STUB1 transfected group were significantly increased compared to the vector control, indicating that STUB1 promotes AR-V7 ubiquitination. We used 5 microgram of AR-V7 antibody to precipitate AR-V7 in the experiments and the results showed that AR-V7 was equally precipitated. It is essential to compare the ubiquitination status between the vector control and the STUB1 transfected groups under an equal amount of AR-V7 precipitation circumstance. Similar techniques were used in other publication (*PNAS*, 2015 112 (28) 8632-8637).